# Pan-African genome demonstrates how population-specific genome graphs improve high-throughput sequencing data analysis

H. Serhat Tetikol [1,2 ✉], Deniz Turgut [1,2], Kubra Narci [1,2], Gungor Budak [1,2], Ozem Kalay[1], Elif Arslan[1], Sinem Demirkaya-Budak [1], Alexey Dolgoborodov[1], Duygu Kabakci-Zorlu[1], Vladimir Semenyuk[1], Amit Jain[1] & Brandi N. Davis-Dusenbery[1]

Graph-based genome reference representations have seen significant development, motivated by the inadequacy of the current human genome reference to represent the diverse genetic information from different human populations and its inability to maintain the same level of accuracy for non-European ancestries. While there have been many efforts to develop computationally efficient graph-based toolkits for NGS read alignment and variant calling, methods to curate genomic variants and subsequently construct genome graphs remain an understudied problem that inevitably determines the effectiveness of the overall bioinformatics pipeline. In this study, we discuss obstacles encountered during graph construction and propose methods for sample selection based on population diversity, graph augmentation with structural variants and resolution of graph reference ambiguity caused by information overload. Moreover, we present the case for iteratively augmenting tailored genome graphs for targeted populations and demonstrate this approach on the whole-genome samples of African ancestry. Our results show that population-specific graphs, as more representative alternatives to linear or generic graph references, can achieve significantly lower read mapping errors and enhanced variant calling sensitivity, in addition to providing the improvements of joint variant calling without the need of computationally intensive post-processing steps.

[1] Seven Bridges Genomics, Charlestown, MA, USA. [2] These authors contributed equally: H. Serhat Tetikol, Deniz Turgut, Kubra Narci, Gungor Budak. ✉email: serhat.tetikol@sevenbridges.com

Next-generation sequencing (NGS) read alignment and variant calling methods rely on the human genome reference[1,2] in order to make sense of the raw data. The validity and effectiveness of these methods are fundamentally determined by the genome reference. The latest version, GRCh38, is derived from a handful of individuals with approximately 70% pertaining to a single individual, and therefore it fails to capture the genetic diversity of the vast majority of human populations[3–5]. This issue has been highlighted by many studies over the past decade[5–10] and various methods for incorporating a wider breadth of genetic information into the reference have been proposed including nucleotide additions and extensions to the current reference[11–13], de novo assembly of raw read data to generate a population-specific consensus sequence[14–16], and graph-based references capable of simultaneously representing multiple diverse populations[17–22]. All of these methods have trade-offs between accuracy, efficiency, and applicability[23]. In addition to developing a suitable data structure and appropriate algorithms to work with it, choosing the appropriate variation information to incorporate into the reference is an important but understudied problem without a straightforward solution[24].

In order to ensure long-term utility and compatibility, especially with large-scale sequencing projects[25–28], a novel genome reference and associated bioinformatics tools should meet the following criteria: (1) accurate representation of diverse genetic information, (2) compatibility with existing methods and standards[29–32], (3) aptness for improvements and other modifications (see Fig. 1A), (4) computational efficiency and scalability, and (5) tailorability for targeted populations and/or applications. In this study, we propose a population-specific graph construction method that meets all of these criteria and compare its utility in NGS read alignment and variant calling to other approaches. We show that a population-specific genome graph can significantly improve both read alignment and variant calling accuracy while being computationally efficient. Moreover, we show that genome graphs can be augmented to further improve the detection power of both short variants (single-nucleotide polymorphisms (SNPs) and insertions and deletions (INDELs)) and structural variants (SVs). We compare our results with those obtained by using joint variant calling[33], which is the state-of-the-art method for genotyping a large number of samples, and show that a graph-based approach provides most of the improvements provided by joint calling with significantly reduced computational requirements.

## Results

**Study design**. In order to measure the utility of a graph-based method and construct a population-specific graph, we use the Seven Bridges GRAF pipeline[20] and benchmark the pipeline on the Illumina sequencing data of the 661 unrelated African super-population samples from the 1000 Genomes Project phase 3, which constitute the most genetically diverse and also the most under-represented super-population with respect to the current human genome reference[15,34,35]. We split the samples into two sets, each containing the same ratio of males/females and the seven populations of African ancestry (ACB, ASW, ESN, GWD, LWK, MSL, YRI) in the 1000 Genomes dataset. The construction set is used for graph reference construction and the benchmarking set is used to measure the performance of various types of graph references. The full list of samples and their respective groups are available in Supplementary Table 14. We categorize graph references into three main types depending on the sources used to construct them.

1. Pan-genome graph: a graph reference that contains information from many populations around the world[20], collected from public databases.

2. Population-specific graph: a graph reference that contains genetic information pertaining to a single population, usually collected from public databases.

3. Population-specific graph with cohort information: a population-specific graph reference that is augmented with a subset of the sample set under study.

In this study, we construct all three types of graphs for the African super-population and compare the alignment and variant calling performance on the benchmarking set. We show that, as the graph reference becomes more tailored to the population (from type 1 to type 3 above), alignment error rate is reduced and variant calling sensitivity is enhanced. To simulate a multi-phase sequencing project (Fig. 1A), the construction set is split into 5 equal-sized cohorts of 104 samples each, leaving 141 samples for the benchmarking set. Initially, a population-specific graph is constructed using the public database gnomAD[36], which is used to process the first cohort, as shown in Fig. 1B. The variant calls on the first cohort are combined with the initial graph to generate the next graph reference which, in turn, is used to process the second cohort. Starting with the construction of Pan-African 1 graph, we also incorporate the high-quality SVs curated by the Human Genome Structural Variation Consortium (HGSVC) using PacBio HiFi sequencing data for ten African ancestry samples in the 1000 Genomes dataset[37].

The graphs produced at each step are used to process the benchmarking set and evaluate the performance. In addition, a pan-genome graph, which is constructed from multiple public resources and contains the genetic information of many populations[20], is used to compare the population-specific graphs to a non-specific graph. Finally, all graph approaches are compared to the standard linear reference-based approach, i.e., BWA-MEM[38] and GATK, to establish a baseline for a reliable comparison with the existing technologies. The linear BWA +GATK pipeline utilizes joint calling and variant quality score recalibration (VQSR) variant filtration that are the recommended methods for genotyping a large number of whole-genome samples[35].

**Population-specific graph construction**. Variant selection for graph references still remains an open question. Previous studies mostly relied on simple heuristics such as filtering with respect to allele frequency (AF), or methods that may not scale well with large and/or missing information[19,24]. Here, we propose a framework that relies on two basic measurements of a given population: Nucleotide diversity within the population and absolute divergence from the current human genome reference, in this case, GRCh38 (see Methods)[39]. Nucleotide diversity measures the average genetic distance between the individuals from the same population while absolute divergence measures how distant the population is from GRCh38. The results in the whole genome for all five super-populations in the 1000 Genomes dataset are given in Fig. 2A. As expected, the African super-population shows the highest diversity and also the largest divergence from GRCh38. Detailed measurements on individual sub-populations are provided in Supplementary Section 2. Next, we calculate the true positive rate (TPR) and false positive rate (FPR) in graph references constructed for each of the five populations after applying a 5% AF cut-off as a function of the number of samples used for graph reference construction. TPR and FPR represent the ratio of variants that are correctly and incorrectly added to the graph, respectively. It is observed that more diverse populations require a higher number of samples to reach the same level of representativeness (Fig. 2B). The calculation is done both theoretically (Expected) by assuming an underlying AF distribution for the given population, and also

**A**

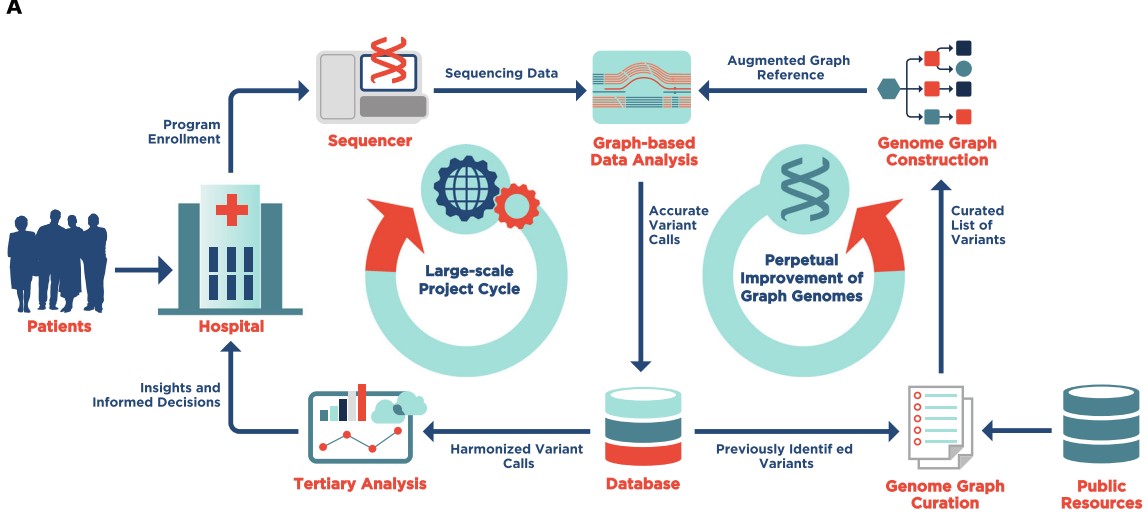

**B**

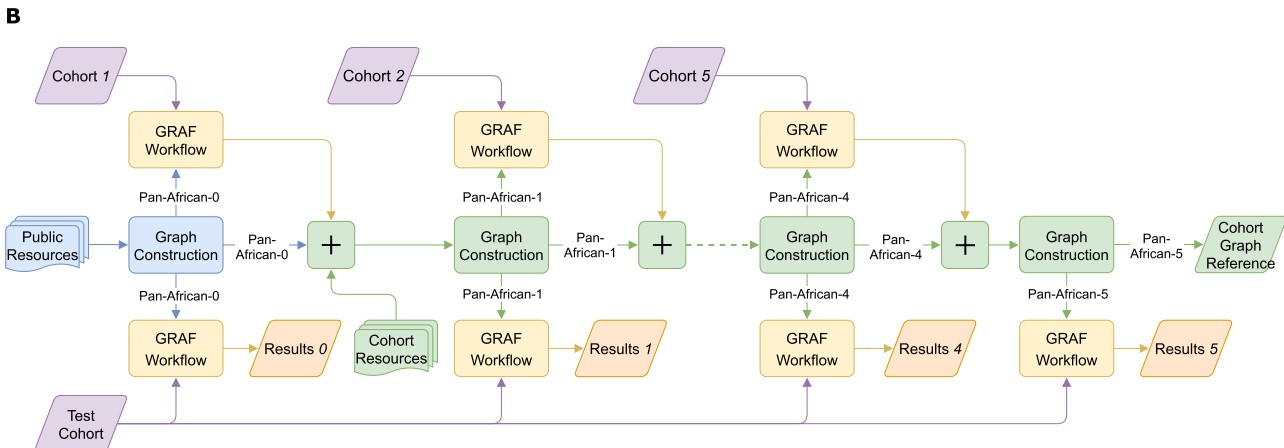

**Fig. 1 Steps involved in a multi-phase sequencing project. A** Large-scale sequencing projects are commonly executed in multiple phases, each comprising the sequencing and bioinformatics analysis of only a subset of the samples that are planned to be sequenced throughout the project (Large-scale Project Cycle). This iterative nature provides the opportunity to produce genomic information in each cycle that can be used to improve the bioinformatics processes (Perpetual Improvement of Graph Genomes). Graph-based secondary analysis approaches can utilize this information to improve the variant detection power for subsequent cycles. **B** Iterative population-specific graph construction workflow. The initial population-specific graph reference (Pan-African 0) is constructed using publicly available variant databases. At each iteration, a subset of the population (construction set) is processed with the current graph, and the variant calls are used to construct the next graph. This process is repeated until the entire construction set is exhausted. All graph references are tested on the same benchmarking set and their performance is evaluated. The population-specific graphs (Pan-African 0-5) are also compared to a generic graph (Pan-Genome) containing genetic information from many populations and to a linear approach using only GRCh38 reference.

empirically (Homogeneous) by picking samples one by one directly from the 1000 Genomes dataset. The details of sample selection, TPR and FPR calculation are provided in the Methods section.

After relevant variants are collected from the cohort samples and other databases, the graph reference is constructed using the pipeline shown in Fig. 2C. The pipeline consists of five main steps. First, all variants are split into biallelics, left normalized, and filtered with respect to user-defined population and quality control criteria. If there are any alternate contigs in the linear assembly with well-defined mappings to the primary chromosomes, they are converted into variants with respect to their mappings and merged with the variants obtained from samples and databases. After the AFs are re-calculated it is processed via a set of filtration steps. The first filtration step, SV Filter, compares the SVs with each other and with the linear assembly/backbone to avoid duplicate sequences and limits the lengths of SVs to avoid computational problems during alignment and variant calling. A

subset of SVs is filtered out as a result of this step. Then, the resultant variant set is used to simulate reads and detect regions that are causing ambiguity in the reference. Identical paths in different regions of the genome graph may cause multi-mapping and reduce the reliability of subsequent variant calling. These regions are resolved by pruning the graph. All filtering steps implement decision-making heuristics that minimize the number of base pairs removed from the graph. The exact details of each step and associated algorithms are provided in the Methods section.

We follow the procedure outlined above to construct graph references at each iteration of the workflow used in the Pan-African experiments (Fig. 1B). Variant counts for each constructed graph and their intersections along with the mean AFs are shown in Fig. 2D. It is seen that the initial iterations capture most of the variation in the population leaving only a small number of relatively low-frequency variants to be discovered in subsequent iterations, agreeing well with the trend in Fig. 2B.

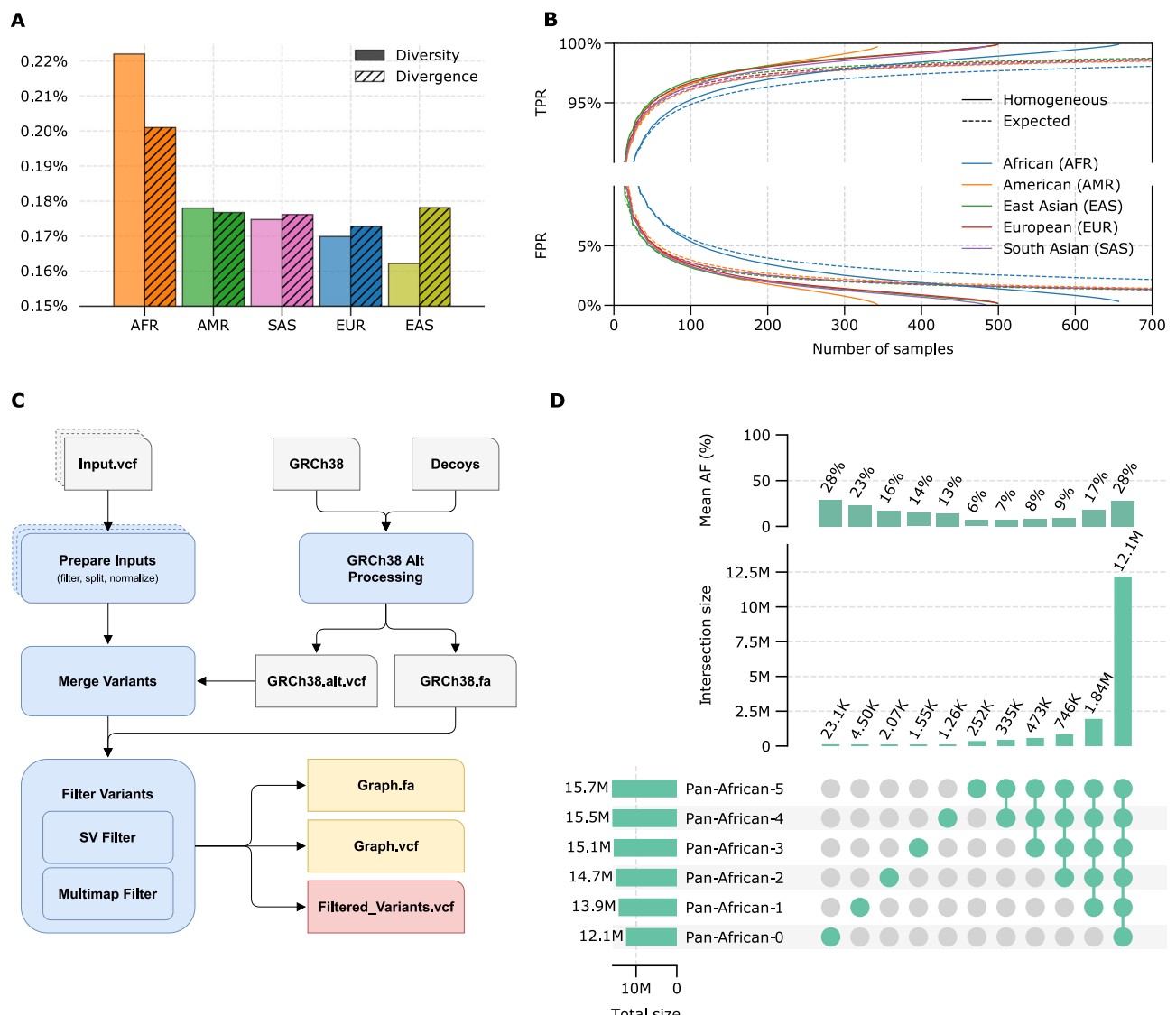

**Fig. 2 Population-specific graph construction summary. A** Nucleotide diversity and divergence with respect to GRCh38 linear reference for each super-population in the 1000 Genomes dataset: African ancestry (AFR), American ancestry (AMR), South-Asian ancestry (SAS), East-Asian ancestry (EAS), European ancestry (EUR). **B** True positive (TPR) and false positive (FPR) rates in the constructed graph references as a function of number of samples used in construction for homogenus (solid lines) and expected (dashed lines) sampling for super-populations; AFR (blue), AMR (orange), EAS (green), EUR (red), SAS (purple). **C** Overview of the graph construction method. **D** Summary statistics for Pan-African graphs constructed at each iteration of the workflow shown in Fig. 1B. Source data are provided as a Source Data file.

A similar analysis on the construction sets are provided in Supplementary Section 1.2.

**Alignment**. The alignment accuracy for each pipeline and/or graph reference is compared as shown in Fig. 3. Figure 3A–F shows various alignment statistics with each violin representing the median and the distribution of the statistic over all benchmarking samples for the corresponding pipeline. BWA maps more reads compared to any of the graph references. This is due to the lenient alignment approach used by BWA as opposed to the more stringent criteria used by the graph aligner. All population-specific graphs map more reads compared to the Pan-Genome approach, while progressively mapping more reads with each graph augmentation step. Improper read (classified as either an improper orientation for read pairs or an insertion length outside the expected range) and uninformative read (MAPQ < 20) percentages are much lower for graph approaches compared to

BWA. Population-specific graphs also provide better performance compared to the Pan-Genome graph. Pan-African 0 graph, although still tailored to the African super-population, is based on public databases and potentially contains many variants irrelevant to the cohort under study. This manifests itself as a larger number of unmapped, improper, and uninformative reads compared to graphs incorporating variants directly obtained from the cohort (Pan-African 1–5).

The multi-mapped read ratio is also higher for BWA compared to any graph approach. A distinct jump is observed between Pan-African iterations 0 and 1. This is due to the addition of cohort-specific SVs into the graph reference. Even though a multi-mapping detection filter is implemented into the graph construction method (Fig. 2C), a graph-based approach will inevitably increase the similarity between genomic regions simply because the genome reference now contains more nucleotide sequences. Our graph construction method effectively balances the trade-off between benefits and multi-mapping as evidenced by

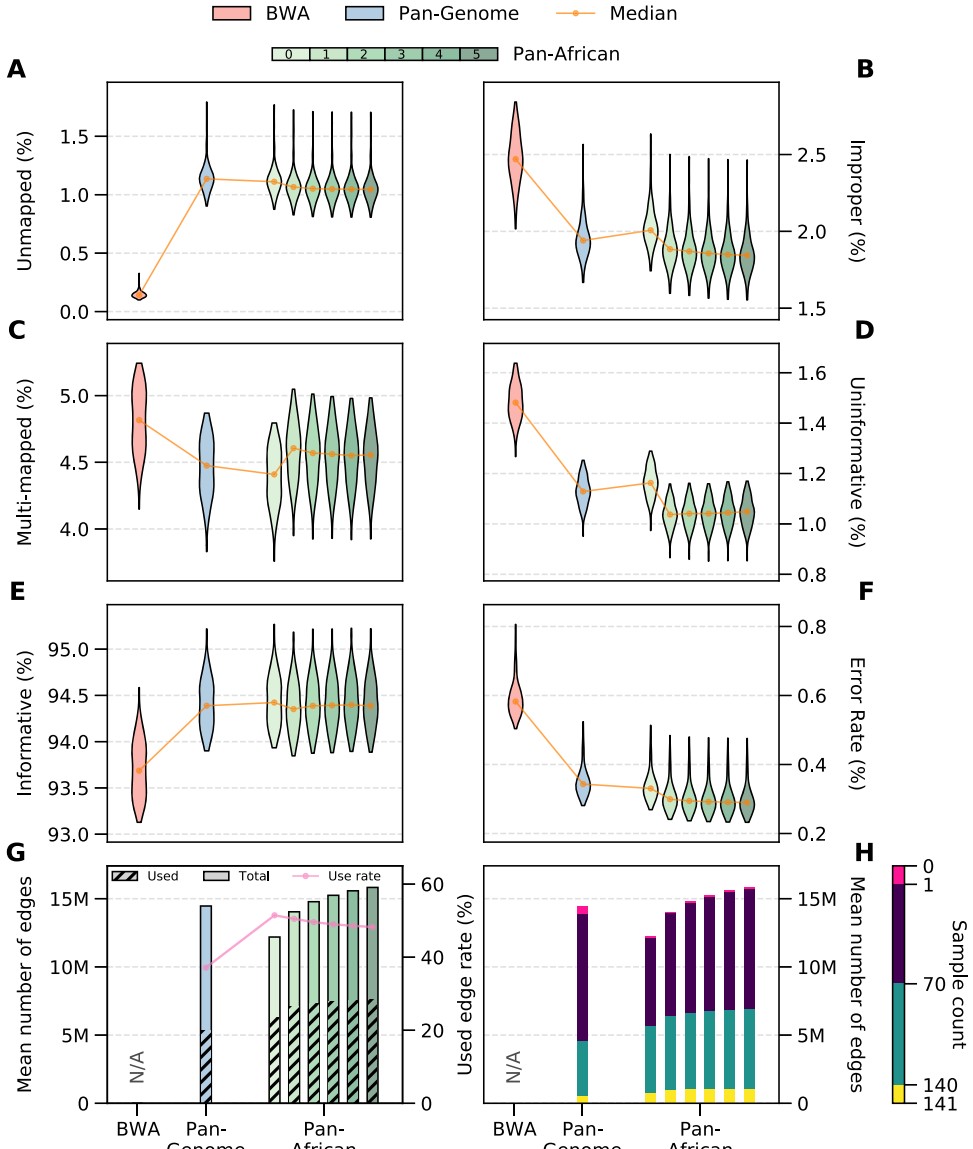

**Fig. 3 Alignment metrics for BWA (red), Pan-Genome (blue), and Pan-African Iterations (green).** Rate of unmapped (**A**), improper (**B**), multi-mapped (MAPQ = 0) (**C**), uninformative (MAPQ < 20) (**D**), and informative reads (MAPQ ≥ 20) (**E**). **F** Alignment error rate. Error rate is the ratio of mismatches to aligned bases in read alignments with respect to the reference. Two-sided Wilcoxon tests between consecutive distributions are performed. In all cases, except for one (uninformative reads between iterations 2 and 3), the difference is significant ($p < 10^{-3}$). **G** Total number of variants in graph (solid bars) and per sample mean of number of used variants/edges in alignment (dashed bars). Magenta line shows the ratio of used variants to the graph size. **H** Categorization of variant utilization in alignment with respect to the number of samples: 0% (pink), below 50% (purple), above 50% (green), 100% (yellow). Source data are provided as a Source Data file.

the reduction in the improper and uninformative read ratios and the alignment error rate from Pan-African 0 to Pan-African 1 (Fig. 3B, D, F).

Graph approaches provide a significantly higher number of informative reads (MAPQ ≥20) as shown in Fig. 3E. The difference in the number of informative reads between the graph types is minimal. However, it is equally important to have the reads align to their proper places in addition to having high mapping quality. In this case, the reads are aligning to the population-specific haplotypes in the graphs and there is significant relocation of reads both within chromosomes and between chromosomes (see Supplementary Section 3.2 for details), which is demonstrated in the next section by the increased sensitivity in variant discovery and SV genotyping.

A useful metric to measure the representativeness of a population-specific is the alignment error rate, i.e. per-base

mismatch rate with respect to the genome reference. A smaller error rate indicates that the genetic composition of the population is more successfully captured and also the reference bias is reduced. Figure 3F shows that the error rate consistently decreases from the linear method BWA to the Pan-Genome graph and to the Pan-African graphs. Each augmentation of the Pan-African graph achieves a better error rate, leading to around 50% reduction compared to BWA. This is an indication of the accuracy improvements that the iterative graph construction approach can provide.

Next, we investigate how much each graph is utilized during read alignment. We calculate the average number of graph edges that are used in alignment per sample and compare it to the total number of edges in the constructed graph reference (Fig. 3G). The ratio of the two is also shown as the magenta line. Population-specific graphs provide a utilization rate of around

50% per sample, whereas it is below 40% for the Pan-Genome graph. Notably, Pan-African 1 graph provides better alignment performance than the Pan-Genome graph even though it is smaller in size. This implies that a targeted graph reference is more beneficial than a more encompassing but generic graph reference, mainly because the irrelevant variants in the generic graph can act as misinformation and cause ambiguity for read alignment. It is also observed that each graph augmentation grows the size of the graph and the number of edges being used, as expected. The utilization rate is slowly reduced with each augmentation. This agrees with the results in Fig. 2D, which shows that the mean AF of the variants added in each subsequent iteration is lower. Figure 3H shows the total utilization of the graph references across all benchmarking samples. All edges are binned by the number of samples that make use of them in alignment, with yellow and magenta bars showing the number of edges used by all and none of the samples, respectively. The Pan-Genome graph has the highest number of unused edges and also the least number of edges used by all samples. The usage of graph edges increases with each augmentation with a trivial amount of unused edges.

**Variant calling**. We show the overall performance of SNPs, INDELs, and SVs for all graph references in Fig. 4. Figure 4A, C shows the number of SNPs and INDELs discovered per sample, respectively. Pan-Genome graph provides a higher sensitivity compared to the BWA+GATK pipeline. Moreover, the Pan-African graphs 0 to 5 can increasingly detect more variants compared to the Pan-Genome graph. Both SNPs and INDELs are annotated using the dbSNP154 variant database[40] and labeled as known or novel. The fraction of known variants is around 87% for all pipelines, which implies that most of the additional variants detected by the graph pipelines are previously discovered in other studies and therefore are likely to constitute a reliable variant call set. Detailed variant counts categorized into genotypes, variant type, and known/novel with respect to dbSNP154 are provided in Supplementary Table 4. SNP Ti/Tv and INDEL het/hom ratios are provided in Supplementary Tables 10 and 12.

Figure 4E shows the number of SVs detected by each pipeline (SVs are defined as variants longer than 50 base pairs). The size distribution of SVs is also shown in Fig. 4F for BWA+GATK, Pan-Genome, Pan-African 5 pipelines. The linear approach BWA +GATK has a significantly lower SV detection rate and can only detect short SVs. The improvement in the pan-genome approach is made possible by the addition of the alt-contigs in the GRCh38 assembly as alternate paths into the graph reference. Starting with Pan-African 1 graph, SVs obtained directly from the 1000 Genomes African ancestry samples are used to augment the population-specific graphs. The result of this augmentation is a much higher rate of SV detection and also an increase in the size of SVs that can be genotyped.

Next, we compare the variant calls made by the Pan-African 5 and the BWA+GATK pipelines in more detail. Figure 4B, D shows the cumulative variant counts for both pipelines with respect to the AF (detailed counts are provided in Supplementary Table 7). The variants are first classified into SNPs and INDELs (Fig. 4B, D, respectively), and then into shared (detected in the benchmarking samples by both pipelines) and unique (detected by either pipeline) variant sets. High concordance is observed between the pipelines as a majority of the variants are detected by both pipelines (solid lines). In order to distinguish between the genotyping efficacy of these methods, shared variants are further split into two categories as $AF_{GRAF} > AF_{GATK}$ and $AF_{GATK} > AF_{GRAF}$ (dotted lines). The former represents the number of variants that are detected in the population by both methods but genotyped with higher sensitivity by the graph pipeline (and by the GATK pipeline, for the latter). Among the variants observed in the population with a high frequency (≥5%), graph pipeline is able to genotype approximately 120k INDELs and 119k SNPs with a higher AF, whereas the same numbers for GATK are 106k INDELs and 51k SNPs. In addition, it is noteworthy that the graph-based approach identifies approximately six times as many unique variants as the linear method.

To measure the putative functional impact of variants, we stratify all variants into coding, intronic and intergenic regions and divide them into three frequency bins as singleton (observed in only one sample), rare (AF < 5% but observed in multiple samples) and common (AF ≥ 5%). Table 1 shows the number of variants unique to each pipeline (detailed results are provided in Supplementary Table 5). The use of Pan-African graph leads to the detection of three to four times more high and moderate impact variants in coding regions for all frequency bins, compared to the BWA+GATK pipeline. The contrast between the linear and the graph approaches increases in intronic and intergenic regions. Similar trends are observed for low impact and modifier mutation events in all regions. These observations indicate that the Pan-African graph improves the sensitivity for both common and rare variants across the whole genome.

Finally, we compare the graph pipeline results to the genotyping improvements facilitated by joint calling in the BWA+GATK pipeline: a method that uses a genomic VCF (GVCF) file obtained from read alignments for each sample independently, and then jointly genotypes all GVCF files together[33]. The result is a multi-sample VCF file containing genotype information for all samples at each loci where there is variation in the population. One of the fundamental advantages of joint calling is that it can recover missing variants and correct genotypes in individuals by considering the rest of the population. The recommended step after joint calling is the VQSR, a machine learning-based variant filtration tool that is effective on a large number of samples[33].

A population-specific graph reference readily captures the population's genetic diversity and therefore it is able to utilize this information not just in variant calling but also in read alignment, unlike the standard joint calling approach. To measure how these two approaches compare to each other, we extract the variants that are recovered/corrected by the BWA+GATK joint calling pipeline, i.e. variants that would have been missed with single sample calling, and look at the concordance of these variant calls with the calls made by the graph-based approach. These recovered variants are further classified by VQSR as PASS and non-PASS, indicating whether or not they pass the filtration criteria to be considered high-quality variant calls. The percentage overlap of these recovered/corrected genomic loci with the calls made by each graph pipeline is shown in Fig. 4G. The Pan-African 5 graph is able to genotype almost 78% of the variants recovered by traditional joint calling (11,763,827 out of 15,088,205 genotypes across all benchmarking samples), without the need of a post-processing step, while calling less than 18% of the variants filtered out by VQSR (1,625,108 out of 9,049,451 genotypes), providing both sensitivity and specificity. Note that the Pan-Genome graph also provides most of the improvements provided by joint calling. The detailed breakdown of variant counts for each genotype and variant type is provided in Supplementary Table 6. The genotype qualities of these variants of further investigated in Supplementary Section 4.2. The results show that the Pan-African graph leads to higher quality genotypes while the variants that are not recovered by the GRAF workflow tend to have lower qualities.

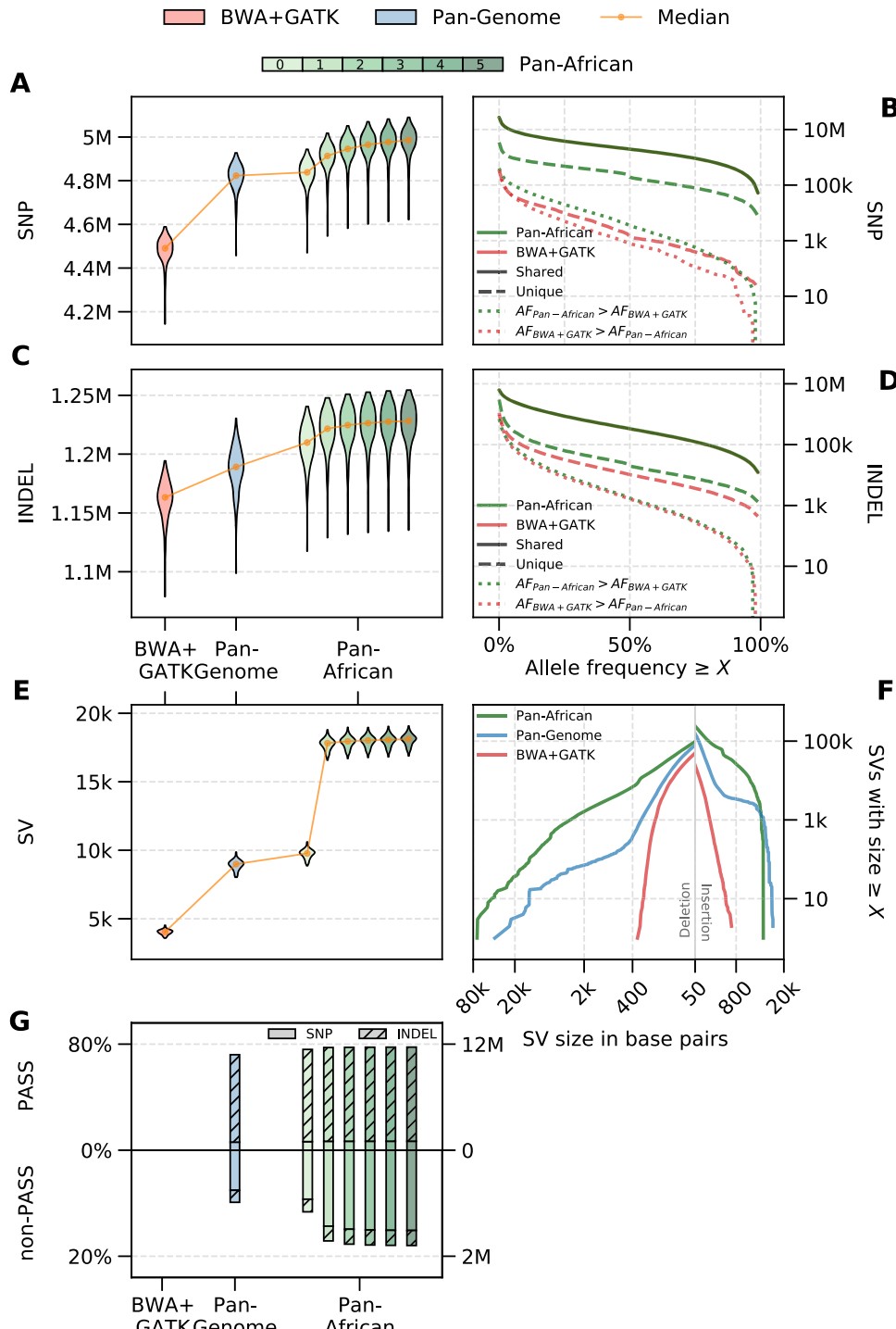

**Fig. 4 Variant calling results for BWA+GATK (red), Pan-Genome (blue), and Pan-African Iterations (green).** **A** Sample distribution of SNP counts, **B** cumulative AF distribution of SNPs separated into shared variants (solid lines), unique variants (dashed lines), and common variants with allele frequency difference (dotted lines), **C** INDEL counts, **D** cumulative AF distribution of INDELs separated into shared variants (solid lines), unique variants (dashed lines) and common variants with allele frequency difference (dotted lines), **E** structural variant (SV) counts, **F** size distribution of detected SVs, and **G** percentage of loci called by the graph pipeline for the variants rescued in the traditional joint calling (results are split based on the filtration output of VQSR). Two-sided Wilcoxon tests between consecutive distributions are performed for **A**, **C**, and **E**. In all cases, the difference is significant ($p < 10^{-21}$). Source data are provided as a Source Data file.

## Discussion

In this study, we have shown that Pan-Genome graphs can provide an overall accuracy improvement over linear references when analyzing NGS samples regardless of their ancestry, and population-specific graphs that are augmented with cohort-specific information provide the highest utility in read alignment and variant calling. The performance of such graphs are ultimately determined by the nucleotide diversity within the population, the absolute divergence from the linear reference sequence, and the availability of pre-existing genomic information to be

**Table 1 Functional impact of detected variants that are unique to each pipeline.**

| Region | Impact | Singleton (graph) | Rare (graph) | Common (graph) | Singleton (GATK) | Rare (GATK) | Common (GATK) |
|--------|--------|-------------------|--------------|----------------|------------------|-------------|---------------|
| Coding | High | 935 | 259 | 145 | 419 | 294 | 197 |
| Coding | Moderate | 5855 | 3582 | 2944 | 1371 | 1028 | 525 |
| Coding | Low | 2533 | 1698 | 1497 | 570 | 404 | 270 |
| Coding | Modifier | 36,781 | 29,140 | 20,203 | 5354 | 6949 | 2841 |
| Intronic | High | 297 | 218 | 158 | 39 | 31 | 17 |
| Intronic | Low | 1310 | 982 | 730 | 159 | 171 | 133 |
| Intronic | Modifier | 1,239,286 | 998,546 | 404,136 | 255,666 | 321,315 | 92,814 |
| Intergenic | Moderate | 458 | 389 | 206 | 100 | 102 | 17 |
| Intergenic | Modifier | 1,575,710 | 1,287,305 | 831,713 | 240,360 | 300,409 | 99,425 |

Variants are split based on their occurrence: singleton (observed in a single sample), rare (AF < 5%), and common (AF ≥ 5%).

used for graph construction. Despite the quantitative similarity, these two metrics are fundamentally independent. Nucleotide diversity measures the average genetic distance between any pair of individuals from the same population and determines the number of samples required to construct a representative graph; Diverse populations will require a larger number of samples for graph construction. In the context of NGS secondary analysis, the reference genome is used only to have an intermediate representation of an individual's genome, therefore it does not influence, at least theoretically, the diversity measurement. On the other hand, absolute divergence is the genetic distance to an arbitrarily defined DNA sequence (e.g., GRCh38), and its value can range from low to high for populations with a similar diversity, as empirically shown in Fig. 2A. For example, the East-Asian super-population shows a large divergence from GRCh38 while showing the least amount of diversity among the five super-populations. Absolute divergence can significantly influence the performance of bioinformatics methods, and standard approaches can suffer from a loss of accuracy when used on divergent populations, which is usually reflected in increased reference bias in alignment and decreased variant calling sensitivity. In instances of large divergence, it may be desirable to additionally incorporate results from orthogonal technologies (e.g., long-read sequencing data[41–43]) to compensate for the shortcomings of the sequencing technologies already used.

The variant curation method for graph construction presented in this study lays out a procedure for sample selection that is based on the population's genetic diversity. The representativeness of the resultant genome graph, along with the required number of samples, is also calculated with associated TPR and FPR as shown in Supplementary Table 1. This table can be used to estimate the number of samples needed for graph construction, while mindfully noting that the size and the sampling of the population ultimately determines how effectively the genome graph converges to the targeted level of representativeness. We expect that the improvements obtained from a genome graph will be more dramatic for divergent populations. This is especially important since it has been well documented that accurate and sensitive detection of variants in under-represented populations plays a critical role in the accuracy of frequently used population genomics methods such as GWAS[44,45]. We have tested the graph construction method on the African super-population which is both the most diverse and the most divergent among the five populations in the 1000 Genome dataset. While exemplifying the suitability of genome graphs for large-scale sequencing projects, the iterative graph construction approach emphasizes the importance of extracting genetic information directly from the cohort under study and making the graph reference more tailored to the cohort. Transitioning from the linear human genome reference GRCh38, which is the least specific reference, to a pan-

genome and finally to a set of population-specific graphs is shown to improve secondary analysis on multiple fronts. We expect this improvement trend to continue as graphs are further tailored to sub-populations or even smaller groups of individuals with higher genetic similarity, assuming the number of samples in the group remains sufficiently large to capture the genetic architecture.

It should be noted that a genome graph representative of the current cohort may eventually show reduced relevance to samples sequenced in the future. This may be caused by reasons such as (i) insufficient sampling during initial construction, (ii) new samples being a part of a previously omitted subpopulation, and (iii) new samples not belonging to the target population, among other possibilities. To facilitate the identification and correction of such issues, the iterative approach enables the detection of an incorrect augmentation, a mismatch between the target population and the analyzed sample, or a quality control problem with the sequencing data by monitoring metrics such as graph utilization directly from the alignments, as shown in Fig. 3. This provides the opportunity to make an informed decision to further augment the graph, prune the existing graph, or construct a second graph dedicated to a new subpopulation, which may prove advantageous especially in large sequencing projects. In order to test the generalizability and the effectiveness of the Pan-African graph reference constructed in this study, we measured its performance on a set of independent African samples from the Human Genome Diversity Project (HGDP)[46]. This dataset contains samples from seven African population groups (Mandenka, Yoruba, Biaka, Mbuti, San, Bantu/Kenya, and Bantu/South Africa; full sample list is provided in Supplementary Table 15), only one of which is captured in the 1000 Genomes dataset. The results show that the Pan-African graph constitutes a more accurate representation of the African populations in the HGDP dataset compared to the Pan-Genome graph or the linear reference GRCh38, and therefore leads to more accurate alignment and more sensitive variant calling (Supplementary Section 1.3 and Supplementary Figs. 6 and 7). These results support the hypothesis that, even though six out of seven African populations were not explicitly used to construct the Pan-African graph, we were able to capture the shared genetic background through other samples of African ancestry and public databases.

There are several pitfalls one might face during the construction of a graph reference. With an aim to address these, our graph construction method takes into account the inadvertent ambiguity introduced to the reference genome as more variants are added and resolves the culpable graph paths. This is a crucial step regardless of the types of variants being added to the graph; we have observed in the construction of Pan-African graphs that even a few SNPs might cause undesirable amounts of read multi-mapping. Moreover, great care should be exercised if large variants such as SVs are being added to the graph to make sure that

they do not contain high similarities to the reference genome or to each other. These control mechanisms lead to an effective graph construction with alignment benefits such as a higher rate of informative reads and reduced reference bias, which remains an obstacle for linear methods, especially in genomic studies of under-represented populations. Since the applicability of our graph construction method is not limited to the specific tools used in this study, we expect that similar improvements can be obtained for other graph-based bioinformatics toolkits.

We have shown that population-specific genome graphs facilitate the detection of more variants and genotyping of SVs at the population scale. We have been able to identify thousands of functionally important variants in the coding regions that are completely missed by the standard BWA+GATK pipeline. In addition, we have detected significantly more previously unknown SNPs and INDELs when compared against dbSNP154 (see Supplementary Table 4). Further studies are required to truly understand the impact of these variants for research and clinical applications. Another advantage of population-specific graphs is that they can readily provide the sensitivity and specificity improvements expected from joint calling without the need for simultaneous processing of all samples in the cohort, where we have also shown that the variants that are genotyped using the Pan-African graph tend to have higher confidence than those that are not genotyped. This capability of a graph-based approach removes the computational burden faced by joint calling when applied to large cohorts.

## Methods

**Nucleotide diversity and absolute divergence.** Graph references rely on enhancing the linear reference with common polymorphisms observed in a given population. Therefore, it is important to estimate the level of polymorphism within the population (diversity) as well as how much the population differs from the linear reference (divergence).

Polymorphism within a population is commonly measured with nucleotide diversity defined by Nei and Li in 1979[39]. This measure calculates the average number of nucleotide differences between two sequences for all possible pairs within a population. Since variant call format defines sequence differences with respect to a common linear reference, nucleotide diversity can simply be calculated from variant calls of samples from a population. For a given variant locus, diversity contribution will be sum of the number of base differences between two alleles weighted with respect to their occurrence frequencies over all possible allele pairs at that loci. Diversity for a genomic region will be the sum of all diversity contributions from variant loci in the region divided by the size of the region.

$$\text{Diversity} = \frac{\sum_{\text{variant loci} \in R} \sum_{i,j \neq i} \frac{|i||j|\delta_{i,j}}{N(N-1)}}{|R|} \quad (1)$$

where $i, j$ are the distinct alleles at given loci, $|i|$ is the number of occurrences for a particular allele, $\delta_{i,j}$ is the edit distance between two alleles, $N$ is the total number of alleles at given loci and $|R|$ is the number of bases within the genomic region.

Divergence is similar to diversity, but instead of measuring the differences between two samples from the population, each sample in the population is compared against the linear reference. Therefore the divergence contribution at a variant locus is average nucleotide differences between an allele and the reference weighted by the occurrence frequencies of those alleles. Divergence within a genomic region is then the sum of all divergence contributions divided by the size of the region.

$$\text{Divergence} = \frac{\sum_{\text{variant loci}} \sum_i \frac{|i|\delta_{i,r}}{N}}{|R|} \quad (2)$$

where $\delta_{i,r}$ is the edit distance between allele $i$ and reference allele at that loci.

**Calculation of TPR and FPR in graph construction.** Graphs can enhance a linear reference by incorporating common variations that can be selected by applying AF thresholds on the population variants. Furthermore, since graphs are constructed from a subset of samples from a population, observed AF of a variant in the subset can differ from the ideal AF in whole population thus resulting in a different set of variants to be selected for graph. Variants in the graph constructed from a subset can be divided into two groups: true variants (ideal AF above cut-off) and false variants (ideal AF below cut-off). TPR is the rate of true variants with respect to ideal graph size, and similarly, FPR is the rate of false variants with respect to ideal graph size.

We investigate the influence of the number of samples on the representativeness of the constructed graph reference and show that it correlates well with the population's nucleotide diversity. In this experiment, we pick samples from a given population one by one and measure how much of the population's genetic variation is correctly captured in the graph reference. To include only common variants in the graph, we use an AF cut-off of 5%, variants below which are discarded. At each step, we compare the content of the graph with the complete population information (obtained from all available samples), label any high (AF ≥ 5%) and low (AF < 5%) frequency variants in the graph as true and false positives, respectively, and finally calculate TPR and FPR. This procedure is repeated for each super-population (AFR, AMR, EAS, EUR, SAS) independently. For result with different AF thresholds and sampling strategies, see Supplementary Section 2.

The TPR and FPR are also calculated theoretically, assuming an underlying AF distribution obtained empirically for each population. Ideal AF of a given variant can be considered as the occurrence probability of that variant for each allele. Assuming the occurrence of variant is independent for each allele, number of times a variant is observed in $N$ diploid samples ($2N$ alleles) follows a binomial distribution with AF as success probability. For an AF cut-off ($f_c$) used in graph construction, the probability of adding a variant with true AF $f$ into graph in $N$ samples is then sum of probabilities where occurrence count results in observed AF larger than cut-off ($k/2N \geq f_c$).

$$P(\text{added}|f) = P\left(\frac{k}{2N} \geq f_c | f\right) = \sum_{k \geq 2Nf_c}^{2N} \binom{2N}{k} f^k (1-f)^{2N-k} \quad (3)$$

TPR can be calculated by the expected fraction of true variants added to graph in $N$ samples divided by the fraction true variants.

$$\text{TPR} = \frac{\int_{f_c}^1 P(\text{added}|f)p(f)df}{\int_{f_c}^1 p(f)df} \quad (4)$$

Similarly, FPR can be calculated by the expected fraction false variants added to graph in $N$ samples divided by the fraction of true variants.

$$\text{FPR} = \frac{\int_0^{f_c} P(\text{added}|f)p(f)df}{\int_{f_c}^1 p(f)df} \quad (5)$$

**Graph construction.** It is common to use the linear reference as the backbone of the graph reference, which facilitates variant representation with respect to the same sequence and the coordinate system, ensuring compatibility with the standard bioinformatics tools[19,20,47,48]. We assume the same approach and use the GRCh38 assembly as the backbone for all graphs. There are the so-called alt-contigs in the GRCh38 assembly, which represent alternate sequences for certain regions in the canonical chromosomes. These regions show high variability in the population and alt-contig haplotypes are provided as additional sequences to augment the haploid genome. However, the natural way of incorporating alternate sequences is indeed adding another path to the graph reference. Therefore, we have developed an alt-contig processing step that removes alt-contigs from the GRCh38 assembly, maps them to the canonical regions, and finally adds them as graph paths with an appropriate representation. First, the contigs labeled as ALT and NOVEL are removed from the linear reference so that it only contains the primary chromosomes, unplaced and unlocalized contigs, and decoy sequences. Next, ALT and NOVEL contigs are mapped to the primary chromosomes. Since they usually contain long stretches of sequences that are identical to the linear reference, ALT and NOVEL contigs are decomposed into smaller variants and left normalized. The final outputs are a modified linear reference that does not contain the alt-contigs and a VCF file that concisely represents alt-contigs with respect to the linear reference.

Additional variations from public sources are collected and harmonized through normalization, splitting multiallelics, and filtering by AF. The use of an AF cut-off is a vital filtering step for graph-based methods and justified by the fact that a low-frequency variant will pose misinformation to most of the samples and make the alignment more ambiguous and genotyping less accurate for those samples. The exact AF cut-off is a free parameter and can be chosen depending on the specific application, population, or type of sequencing data. We have observed that a value of 5% provides shows good performance without incurring large computational costs (see Supplementary Section 1.2). Here, we used gnomAD database with an African AF cut-off of 5% in addition to structural variations observed in African ancestry samples from the HGSVC dataset. After each construction iteration, the resulting variant calls above the 5% AF cut-off are used to augment the graph.

After the variant sources are harmonized, they are merged and final allele frequencies are calculated. Given a large set of variant sources, adding all variants to the graph structure could cause computational inefficiencies, and also lead to inaccuracies in alignment and variant calling. Therefore, the resultant merged VCF file is passed through a couple of filtration steps to remove variants that could cause issues. The first step is the filtration of SVs, which processes all SVs to resolve any ambiguity that might be introduced to the reference due to the nature of short-read sequencing data. First, the insertion SVs are aligned to the linear reference and any SV with an identical subsequence of at least read length (150 bp) is filtered.

Next, the remaining SVs are aligned to each other and if similarities of at least read length are identified, shorter SVs are filtered. Finally, the remaining SVs are aligned to the decoy sequences in the linear reference and if a match is found, matching sections on the decoys are masked with N-bases in order to prevent these sections from acting as sinks for valid reads on these SVs.

As more variants are added, the number of possible paths in the graph grows exponentially and it becomes highly likely that there will be identical paths in different regions of the graph. This issue introduces ambiguity to the genome during read alignment, potentially causing sequencing reads to be multi-mapped and become uninformative for variant calling. Therefore, the next step is a multimap filter that breaks the identity of such paths in the graph by selectively removing variants from it. This is achieved by simulating reads that traverse all possible paths in the graph reference, mapping these reads back to the graph reference, identifying regions that cause multi-mapping, and pruning the variants in these regions of the graph. We calculate the smallest set of variants that resolves the ambiguity and only remove those variants to avoid detracting from the representatives of the graph reference.

A more detailed explanation of the graph construction pipeline is provided in Supplementary Section 1.1. The detailed breakdown of each graph used in this study into variant types can be found in Supplementary Table 2. The contribution of each variant source to the final population-specific graph Pan-African 5 is given in Supplementary Table 3.

**Reporting summary**. Further information on research design is available in the Nature Research Reporting Summary linked to this article.

## Data availability

The public high coverage dataset from 1000 Genomes project is available at https://www.internationalgenome.org/data-portal/data-collection/30x-grch38. The public structural variation dataset from Human Genome Structural Variation Consortium is available at https://www.internationalgenome.org/data-portal/data-collection/hgsvc2. The public gnomAD v3 dataset is available at https://gnomad.broadinstitute.org/downloads. The public sequencing sample dataset from the Human Genome Diversity Project is available at https://www.internationalgenome.org/data-portal/data-collection/hgdp. The Genome in a Bottle benchmarking samples are available at https://www.nist.gov/programs-projects/genome-bottle. The linear reference is based on GRCh38 patch 13 (available at https://www.ncbi.nlm.nih.gov/assembly/GCF_000001405.39) with decoy sequences hs38d1 (available at https://www.ncbi.nlm.nih.gov/assembly/GCA_000786075.2) and Epstein-Barr virus (available at https://www.ncbi.nlm.nih.gov/assembly/GCF_002402265.1). The Pan-Genome and Pan-African graph references are immediately available to all academic researchers through the public files repository of the NHLBI BioData Catalyst and can be made available on request for academic researchers on all other Seven Bridges cloud platforms (restrictions apply for commercial use, please contact Seven Bridges for terms and other details). To access the Pan-Genome and Pan-African graphs on the NHLBI BioData Catalyst, researchers must make a free account using their eRA Commons credentials (available through the US National Institute of Health) and then navigate to the public file repository at https://platform.sb.biodatacatalyst.nhlbi.nih.gov/resources/public-gallery/files. Any academic researchers who do not wish to access the files through the NHLBI BioData Catalyst may email support@sevenbridges.com and our 24/7 helpdesk team will respond with a target response time under 24 h with an alternative access approach tailored to the user's request. Requestors will be asked to verify that their use is for academic purposes only. Source Data are provided with this paper.

## Code availability

The tools and methods presented in this study are available on all Seven Bridges cloud platforms. The graph-based sequencing data analysis tools used in this study (Seven Bridges GRAF) are freely available to all researchers on all Seven Bridges academic cloud platforms (Cancer Genomics Cloud, Cavatica, BioData Catalyst). To access academic platforms, please reach out via the respective website. The source code of the graph reference construction method is not publicly available. Restrictions apply for commercial use, please contact Seven Bridges for terms and other details.

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

## Acknowledgements

We thank our colleagues at Seven Bridges Genomics Inc. for their contributions. In addition, we are grateful for support for the academic platforms that enable researchers to quickly access data and tools described in this study. The BioData Catalyst powered by Seven Bridges is supported by OT3 HL142478-01. The authors wish to acknowledge the contributions of the consortium working on the development of the NHLBI BioData Catalyst ecosystem. The Cancer Genomics Cloud, powered by Seven Bridges, is a component of the NCI Cancer Research Data Commons and has been funded in whole or in part with Federal funds from the National Cancer Institute, National Institutes of Health, Department of Health and Human Services, under Contract No. HHSN261201400008C and ID/IQ Agreement No. 17X146 under Contract No. HHSN261201500003I. CAVATICA is a data analysis and sharing platform designed to accelerate discovery in a scalable, cloud-based compute environment funded in whole or in part by U2CHL138346, OT2OD030162, and U2CHL156291.

## Author contributions

H.S.T., D.T., and B.N.D. conceived the idea. H.S.T. supervised the overall study. D.T. and H.S.T. invented the graph construction methodology. H.S.T., D.T., K.N., and G.B. designed the experiments, implemented the bioinformatics workflows, performed the experiments, analyzed the results, and wrote the manuscript. A.D., V.S., D.K., and A.J. provided support on the graph-based tools. O.K., E.A. and S.D. provided support on benchmarking measurements. All authors discussed and provided feedback on the results and the manuscript.

## Competing interests

All authors have been employed by Seven Bridges Genomics Inc. during this study.
