## [Peer Review File · Nature Communications]

Reviewers' Comments:

Reviewer #1:

Remarks to the Author:

This explores the creation of sub-population-specific pangenome references, using genomes of African ancestry from the 1000 Genomes project to explore the topic. Overall the paper is clearly written, tackles a contemporary topic of some importance, and makes some interesting contributions. However, we have a number of significant concerns as it stands:

- In writing a paper on "Population-specific genome graphs", the authors need to pin down what they mean by "population". 1000GP uses a bunch of individual "populations", like "LWK" AKA "Luhya in Webuye, Kenya", with fairly strict ascertainment criteria on this ancestry. Then it groups them into "superpopulations", like in this case "AFR" AKA "African Ancestry". The paper here uses the word "population" throughout for everything; it talks about "the 7 African populations in the 1000 Genomes dataset", and then also about how "the East Asian population shows... the least amount of diversity among the five populations". When they mean "superpopulation", they need to say "superpopulation". Otherwise when the methods says "This procedure is repeated for each population independently." it's completely ambiguous what they are iterating over.

- Notably "AFR" as a superpopulation also includes "African Caribbean in Barbados" and "African Ancestry in Southwest US"; these are somewhat oddly included in what the paper calls "the 7 African populations in the 1000 Genomes dataset". In our view, there's a reason why 1000GP uses the more unwieldy "African ancestry" descriptor. The authors here should consider using the terminology recommended by 1000 Genomes and the people who collected the samples.

- The paper's "Pan-Genome" non-population-specific graph looks to be just the author's previous graph from reference 20, "Fast and accurate genomic analyses using genome graphs". We think it used a completely different construction procedure, and not the combination of gnomAD, GRCh38 alt loci, and iterative improvement with more samples as is detailed here for the "Pan-African N" graphs. In particular, we don't think the "Pan-Genome" graph has any of the anti-ambiguity filtering. This is very problematic, because you can't draw a conclusion that a population-specific graph would outperform a similar non-population-specific graph from the experiments presented here. The control the authors need is a graph built with the same procedure as the "Pan-African 0" and "Pan-African 1-5" graphs, but with the African-ness constraints removed; the authors need to use gnomAD global allele frequencies to make a matched "Pan-Genome 0", add in HGVC SVs across all populations for a "Pan-Genome 1", and iteratively augment the graph with 5 construction cohorts of 104 samples each sex- and population-balanced across the whole 1000GP dataset. Without this more careful control the authors are conflating their new construction approach, database set, and filtering heuristics with the benefits of matching the population distribution of your test set.

- We believe they have a useful online approach to building a graph with the right variants from a sample data, but in general we don't think they've evaluated it rigorously enough. For one thing, their sample selection seems circular, in two ways.

(1) The authors split the 1000GP AFR superpopulation into construction and benchmarking sets, but only 141 of the samples they have are in the benchmarking set, and 104×5 of them are in the construction sets. 1000GP has a lot of trios, and they don't say anything about dropping related samples, so by the end of the construction sets it's very likely that benchmarking set samples have close relatives in the graph. If this is not the case it needs to be very clearly stated.

(2) 1000 Genomes populations are special beasts, e.g. "people who claim to have four Yoruba grandparents". Even if you look at a whole superpopulation the data set is still made up of people drawn from these weirdly peaked distributions over all possible ancestries. I.e. they're building a graph with good coverage of exactly the specific individual constrained populations that all the 1000GP populations are. Their evaluation samples are constrained to land exactly atop their construction samples in a way that won't be true for arbitrary people sampled off the street, even if the street is in Africa. The whole thing might fail without this super-peaked distribution of samples. To solve this issue convincingly the paper really needs independent samples that might reflect what you'd get if you selected people who are not so clearly related to African ancestry

subpopulations sampled by 1000 Genomes. Put another way, this is in some ways a best-case scenario. One imperfect way around this would be to throw out, swap out or add populations or individuals to the set, can they still see the benefit of a "population-specific" (i.e. test-set-allele-frequency-distribution-matched) graph?

- Closely, related to the above issue of sample selection, the paper here relies on comparing people with known ancestry. In the real world, only a fraction of people can claim such homogenous ancestry. Further, ethnic and cultural identity and genetics are known to have a complex and hard to predict relationship. The authors should discuss this issue. Further, how would the authors propose to deal with recently admixed individuals? It feels like the solution proposed directly here would only work for a fraction of people, and this limitation should be discussed.

- We think the AF-based metric makes sense for evaluating how well the graph built matches the graph that was intended to be built. But the "AF_GRAF > AF_GATK" statistics, which the authors try to use to show increased sensitivity aren't good metrics, because they're treating every positive as a true positive: the authors can not know what the real AF actually is, so they can't say identifying more copies of the allele means their tool is more sensitive.

Minor points:

- Their title brings in a concept of "the Pan-African genome" which is never mentioned again; Do they mean "African Pan-Genome" or "Pan-African graph" maybe?

- We see where the authors are coming from with the AF accuracy to tell how much a subsample represents the population. But it seems pretty obvious, and not worth so much main text emphasis. Likewise for the diversity / divergence discussion. They say they are "fundamentally independent" but high diversity will generally imply high divergence. (Notably, the paper does not mention that the majority contributor to the existing human reference is a mixrace African-American individual)

- The authors come up with 651 samples, but link to a data set with 893 AFR samples, and 703 AFR samples from populations collected in Africa. Please explain the discrepancy.

- The title brings in a concept of "the Pan-African genome" which is never mentioned again; Do they mean "African Pan-Genome" or "Pan-African graph" maybe?

Reviewer #2:

Remarks to the Author:

The paper manuscript details the generation of increasingly refined graph references to demonstrate population-specific graph references capture more variant information than standard methods such as the GATK/BWA 'best practices' approaches. This include a demonstration that graph references show promise in effectively replacing both joint genotyping and VQSR filtering. In particular, the inclusion of defined metrics to evaluate how particular populations can be assessed is a strong asset that could potentially be applied to other similar analyses in the field.

The manuscript is well-written and analyses are very thorough, and I find the work demonstrates significant promise for the field in pointing to a potential path towards more accurate and sensitive variant calling. I do have some points that should be addressed, though I think they are fairly minor hopefully easily addressable; these are largely outlined below. I am attaching also my notes from the PDF version of the main manuscript which has associated comments.

* I may have misunderstood this point so apologies in advance, but I couldn't find the explanation in the text. In the supplemental methods, specific SVs (insertions) were removed due to the length of the insertion or its similarity to other regions of the genome, thus increasing the potential for ambiguously mapped reads. In addition the multimap filter further prunes the graph based on

current haplotypes and mapping. However, one could argue by leaving these out this may lead to some ascertainment bias while tailoring the graph to a population: short reads that *should* be detected as multi-mapping to a graph won't since the offending insertions are filtered out, so perhaps including these regions and leaving these as multi mapped reads is for the best so they can be detected and assessed accordingly. Can you address this point?

* The alignment rates noted and the methods within the main text don't mention the methods used for alignment to the graph in the main text, but it appears this was attained using the GRAF workflow (per the supplemental). It's worth noting this at least once in the main text.

* It should be clarified early on in the main text (intro) that reference genome graphs are generated from already established variant information (e.g. not from de novo variant calls). This *is* mentioned in the methods and in some of the figures, but I think this would best be made explicit early on in the intro to make this clear up-front

* Were any general comparisons to other graph-based variant analysis approaches performed to determine whether the same trends exist, such as comparisons to vg? If not, could you clarify why?

* The methods suggest that SVs from 1000g PacBio data were added *after* the construction of PanAfrican genome 0. It is noted that the jump from PAG 0 to PAG1 is likely due to this, but is there any reasoning these SVs weren't added in the initial graph construction step (PAG 0)?

* The breakdown of the numbers here is a little confusing; can this be clarified? For example, what about the 22% of variants not recovered compared to joint calling?

> (pg 5) The Pan-African 5 graph is able to genotype almost 78% of the variants recovered by traditional joint calling (11,763,827 out of 15,088,205 genotypes across all benchmarking samples), without the need of a post-processing step, while calling less than 18% of the variants filtered out by VQSR (1,625,108 out of 9,049,451 genotypes), providing both sensitivity and specificity.

* Do you recommend using the approach for graph generation in the manuscript, or is this meant to be more of a proof-of-concept demonstrating that pop-specific graphs are much better at capturing variation? For example, IMO the graph construction methods seem very elaborate, suggesting that regeneration of a graph based on varying the AF cutoff, length and remapping filtering of SVs, etc could be limiting factors in using this approach; re-generating graphs could be computationally challenging and expensive if filtering steps or criteria for inclusion of samples are adjusted, whereas a more comprehensive/inclusive approach that can be post-filtered (or masked) ad hoc might be better.

Reviewer #3:

Remarks to the Author:

NCOMMS-21-34197-T

Population-specific genome graphs improve high-throughput sequencing data analysis: A case study on the Pan-African genome

General comments

Authors have addressed an important question: The bioinformatic methods i.e. Genome Graphs, to improved genome analysis in order to alleviate the disparity in populations variants representativity in the current reference genomes, with a specific focus on African populations.

The study is well designed, executed, and written to accommodate a wide range of specialist and non-specialist readers. The article provides some evidence confirming or supporting that : 1- At the genomic level, African populations is both more diverse and divergent from the current reference genome; 2- More diverse the population is, higher is the number of samples needed to

read an appropriate level of representativity; 3- Population specific progressive graph mapping performed much better, compare to pan genome approach; 4- tailoring genome graph to specific cohorts improved graph construction, alignment and variant calling outcomes; 5- the higher the number of informative read, and the lower the number of alignment error rate... The article provide data suggesting the Genome graphs approach will add, and likely improve the currently available bioinformatics tools kit, while improving analytics for all populations, and particularly the underrepresented and more diverse African populations.

While the article will add to the global literature, it could be improved in a few areas.

Specific comments

1- Regarding Accuracy, Efficiency, and Applicability: Can the author uses their approach to compare/ replicate with in more diverse set of African genome data, such the one recently reported by the H3Africa Consortium?
Nature, 2020 Oct;586(7831):741-748; doi: 10.1038/s41586-020-2859-7. PMID: PMC7759466

Specifically, using the H3Africa genome data, to explore comparatively, using Genome Graphs:

- capturing the number of novel SNV...
- investigating the proportions o of actionable genes variants
- numbering of novel SNVs?
- Etc.....

2- How good can the Genomes Graphs method better characterizes the so-call "Ghost DNA" in African genomes? Can the author provide some evidence?

3- Can the approach be used to identified marker of natural selection, known loci such as Sickle cell disease mutation, alpha-thalassemia, G6PD, APOL 1 variant, or other novel loci under natural selection, that expected to be more common in African genomes?

4- Can the authors comment on the Number of African genomes needed to accurately capture the variations African populations?

Reviewer #1 (Expertise: genomics, variation, bioinformatics):

This explores the creation of sub-population-specific pangenome references, using genomes of African ancestry from the 1000 Genomes project to explore the topic. Overall the paper is clearly written, tackles a contemporary topic of some importance, and makes some interesting contributions. However, we have a number of significant concerns as it stands:

We thank the reviewer for their valuable comments. We have addressed all questions and concerns one by one below, in green color. Where applicable, we have noted the changes we made to the manuscript in light of the reviewer's suggestions.

Comment #1

- In writing a paper on "Population-specific genome graphs", the authors need to pin down what they mean by "population". 1000GP uses a bunch of individual "populations", like "LWK" AKA "Luhya in Webuye, Kenya", with fairly strict ascertainment criteria on this ancestry. Then it groups them into "superpopulations", like in this case "AFR" AKA "African Ancestry". The paper here uses the word "population" throughout for everything; it talks about "the 7 African populations in the 1000 Genomes dataset", and then also about how "the East Asian population shows... the least amount of diversity among the five populations". When they mean "superpopulation", they need to say "superpopulation". Otherwise when the methods says "This procedure is repeated for each population independently." it's completely ambiguous what they are iterating over.

We thank the reviewer for their observation of this issue and agree that there is ambiguity in the way that these terms are currently used in the manuscript. We have now clarified in the manuscript what exactly is meant by these terms. Specifically, we chose to use "super-population" when we refer to AFR, EUR, SAS, EAS and AMR ancestry groups in the 1000 Genomes dataset, and "population" when referring to any subset of these super-populations such as LWK. We added the necessary identifiers to the manuscript where relevant to improve comprehensibility, including the sentences the reviewer found ambiguous. However, we have chosen to still refer to the African genome graphs we construct as "population-specific" graph references. In this context, we use a more relaxed definition for population: A collection of individuals with some putatively shared genetic background. This relaxed definition is important because our graphs (or also some other graph-based toolkits) do not necessarily discriminate between different levels of population; it offers the same improvements with increasing benefits for more targeted groups.

Comment #2

- Notably "AFR" as a superpopulation also includes "African Caribbean in Barbados" and "African Ancestry in Southwest US"; these are somewhat oddly included in what the paper calls "the 7 African populations in the 1000 Genomes dataset". In our view, there's a reason why 1000GP uses the more unwieldy "African ancestry" descriptor. The authors here should consider using the terminology recommended by 1000 Genomes and the people who collected the samples.

As explained in our response to reviewer's first comment, we switched to using the term "super-population" to refer to the collection of populations of African ancestry, i.e. AFR, in the 1000 Genomes dataset. We have also rephrased the sentence the reviewer quotes as "the 7 populations of African ancestry (ACB, ASW, ESN, GWD, LWK, MSL, YRI) in the 1000 Genomes dataset".

On a technical note, it is important to capture all genomic information that is relevant to the African ancestry in the graph reference, independent of where each individual population is located geographically.

Comment #3

- The paper's "Pan-Genome" non-population-specific graph looks to be just the author's previous graph from reference 20, "Fast and accurate genomic analyses using genome graphs". We think it used a completely different construction procedure, and not the combination of gnomAD, GRCh38 alt loci, and iterative improvement with more samples as is detailed here for the "Pan-African N" graphs. In particular, we don't think the "Pan-Genome" graph has any of the anti-ambiguity filtering. This is very problematic, because you can't draw a conclusion that a population-specific graph would outperform a similar non-population-specific graph from the experiments presented here. The control the authors need is a graph built with the same procedure as the "Pan-African 0" and "Pan-African 1-5" graphs, but with the African-ness constraints removed; the authors need to use gnomAD global allele frequencies to make a matched "Pan-Genome 0", add in HGSVc SVs across all populations for a "Pan-Genome 1", and iteratively augment the graph with 5 construction cohorts of 104 samples each sex- and population-balanced across the whole 1000GP dataset. Without this more careful control the authors are conflating their new construction approach, database set, and filtering heuristics with the benefits of matching the population distribution of your test set.

In the construction of our Pan-Genome graph reference (and also any other graph used in this study), we have used the new graph construction approach proposed in this manuscript. Therefore, it includes all the anti-ambiguity filtering steps and the processing of GRCh38 alt-contigs. We cite our previous work, first, to point the readers to the relevant publication if they are interested in the standard benchmarking results commonly used for bioinformatics tool comparison, and secondly, to refer to a discussion around Pan-genome graphs and the genomic variants that go into one.

The reviewer also suggests that we use gnomAD in the Pan-Genome graph. Unfortunately, while gnomAD is appropriate for a population-specific graph, it would lead to biased construction of a Pan-Genome graph because gnomAD contains disproportionate numbers of samples from each ancestry it covers. Please see the table below.

Population	Description	Genomes
afr	African/African American	20,744
ami	Amish	456
amr	Latino/Admixed American	7,647
asj	Ashkenazi Jewish	1,736
eas	East Asian	2,604
fin	Finnish	5,316
nfe	Non-Finnish European	34,029
mid	Middle Eastern	158
sas	South Asian	2,419
oth	Other (population not assigned)	1,047

It is clear that a pan-genome graph made from global AFs from gnomAD will in fact be an African+European graph with some “noise” added from other populations around the world. Trying to normalize AFs with the population sizes would lead to uneven uncertainties for each group. One would be better off just using the relevant ancestry from gnomAD to start their graph reference, which is what we do in our study. On the other hand, we made sure to include all populations proportionally in our Pan-genome graph by starting from sequencing reads, circumventing this issue with gnomAD. Nevertheless, to address any concerns the reviewer might still have, we have constructed a Pan-Genome graph from gnomAD and measured the performance on the benchmarking samples. We attach our results here and also to the supplementary material (at the end of Section S1.3). As seen in the figures below (for descriptions, please refer to the main text figures as the format is the same), the gnomAD pangenome graph’s performance is slightly better than the original pan-genome. This is mainly due to the fact that the gnomAD pan-genome graph is actually an African+European graph. However, it is still significantly worse than population-specific graphs, supporting the claim that population-specific graphs augmented with the cohort at hand provide better utility.

We respectfully disagree with the reviewer's suggestion that we should use the iterative approach also for the Pan-genome graph. Although the reviewer's proposal generates a more ideal comparison from a purely theoretical point of view, it would be an experimental scenario that creates synthetic and sterile conditions that are unlikely to be realized in real life applications.

First, in most cases, it is difficult or impossible to get access to raw sequencing data for Pan-genome graph construction. The most significant example of this is gnomAD, which we used in our study but have no way of iteratively re-processing the individual samples. Other examples are dbSNP, HapMap, dbVar, ClinVar, COSMIC, among many others that also include data for targeted applications (e.g. population, disease). In our Pan-genome graph and our population-specific graph Pan-African 0, we solely rely on publicly available data (not obtained from the cohort under study) to emulate a real-life scenario and provide scientific evidence to guide the decisions one would need to make in a genome sequencing project. We specifically call the Type 3 graph (see the beginning of the Results section) "augmented with a subset of the sample set under study" for this reason. In other words, we start with a population-specific graph made out of publicly available datasets and start augmenting it using the cohort samples which is always available for the person conducting the sequencing project. Please also see the visualization below, comparing the genome graph constructed in our study in terms of population coverage.

Second, it is unfeasible (prohibitively costly and time-consuming) to iteratively re-process all samples for the Pan-genome graph construction. For example, gnomAD contains 76,156 genomes that would cost millions of dollars to reprocess. The reviewer suggests starting with the available gnomAD variants and iteratively augmenting it with the 1000 Genome samples to construct Pan-genome 1-5 graphs; however that would defeat the purpose of reviewer's suggestion. Iteratively processing five cohorts of 104 samples from 1000 Genomes will have little to no contribution when combined with the 76,156 samples from gnomAD for pan-genome graph construction. Since we are talking about a Pan-genome graph here, one will need to re-calculate AFs after each iteration when merging gnomAD and 1000G iteration variant calls. At a maximum (the extreme case where gnomAD has 1 variant at 0% AF and 1000GP has the

same variant at 100%), the global AFs can at most change by approximately 1 part in 1000 for each iteration (on average, this change will be much smaller per variant). This will have only a trivial effect on the graphs constructed.

The reviewer also suggests that we should use the HGSC structural variants in the pan-genome graph. The HGSC variants come from a sample set that is too small (64 samples only) for pan-genome graph purposes. Any structural variant set that goes into a pan-genome graph should come from a consensus call set obtained from a large set of samples or a harmonization of multiple SV studies to cover many ancestries. We use HGSC SVs in the population-specific graphs starting with Pan-African 1 because they are cohort-specific (they come from a small subset of 1000GP samples of African ancestry) and therefore have immediate relevance to the samples under study. This way, our approach enables the genotyping of these SVs and slight variations of them across the entire benchmarking set.

Comment #4

- We believe they have a useful online approach to building a graph with the right variants from a sample data, but in general we don't think they've evaluated it rigorously enough. For one thing, their sample selection seems circular, in two ways.

(1) The authors split the 1000GP AFR superpopulation into construction and benchmarking sets, but only 141 of the samples they have are in the benchmarking set, and 104*5 of them are in the construction sets. 1000GP has a lot of trios, and they don't say anything about dropping related samples, so by the end of the construction sets it's very likely that benchmarking set samples have close relatives in the graph. If this is not the case it needs to be very clearly stated.

(2) 1000 Genomes populations are special beasts, e.g. "people who claim to have four Yoruba grandparents". Even if you look at a whole superpopulation the data set is still made up of people drawn from these weirdly peaked distributions over all possible ancestries. I.e. they're building a graph with good coverage of exactly the specific individual constrained populations that all the 1000GP populations are. Their evaluation samples are constrained to land exactly atop their construction samples in a way that won't be true for arbitrary people sampled off the street, even if the street is in Africa. The whole thing might fail without this super-peaked distribution of samples. To solve this issue convincingly the paper really needs independent samples that might reflect what you'd get if you selected people who are not so clearly related to African ancestry subpopulations sampled by 1000 Genomes. Put another way, this is in some ways a best-case scenario. One imperfect way around this would be to throw out, swap out or add populations or individuals to the set, can they still see the benefit of a "population-specific" (i.e. test-set-allele-frequency-distribution-matched) graph?

(1) In order to have a meaningful change in the composition of the graph references constructed after each iteration (Pan-african 1-5) with sufficiently high confidence around allele frequencies, one needs to use a relatively high number of samples. We have chosen to have a resolution of at least 1% (for any type of variant), hence a number of samples larger than 100 for each iteration. The exact number came out to be 104 samples when we designed the experiments to

capture with balance each population of African ancestry and sex. The African super-population also constitutes the largest group in the 1000GP among other super-populations. Therefore, this is a valid experimental design given the size of the datasets available to us.

We indeed only use unrelated samples in the 1000GP, i.e. we never use related samples (the trios) in either graph construction or benchmarking (or elsewhere). It is an oversight on our part that we did not make this explicit in the manuscript. We clarified this in the first sentence of the Results section. For interested readers, we had already provided the complete sample list used for each graph and benchmarking in the Supplementary Table S14. We now refer to this list in the Results section.

(2) We have measured the influence of completely leaving out distinct African populations on the constructed graph reference. We did this experiment for all five super-populations in the 1000GP dataset. Please see Supplementary Section 2 (discussion around homogeneous and clustered sampling) and Supplementary Figures S7 and S8. We observed that the representativeness of the graph suffers and its convergence rate towards better representativeness slows down. In the extreme case of Admixed American populations (AMR), each AMR population is so drastically different from the others that the representativeness of the graph even drops, but eventually recovers as more samples are added.

These observations are not surprising at all. The use of population-specific genome graphs requires the researcher to capture, as best as possible, the genetic background of their samples in the graph reference by using a subset of the cohort (in addition to other public resources). In the Discussion section, we discuss the recommended course of action for cases where one encounters a sample that is not properly captured by the graph reference (this section has now been expanded) and provide robust graph-based metrics for measuring the drop in performance. Obviously, a graph reference will never be perfect; there will always be novel genetic content discovered in future studies. This is exactly the reason why we are proposing a growing reference structure that learns with new findings as opposed to a static one which is used today and poses many difficulties and compatibility issues when used on under-represented populations or during updates/corrections (e.g. GRCh37 to GRCh38). The goal we are trying to achieve with the Pan-African graph reference constructed in this study is to, first, provide a better starting point for sequencing data analysis on African populations and, second and more importantly, provide a method for constructing a better reference for any future study on any population as new data becomes available, enabling the capture of higher representativeness and more accurate sequencing data analysis.

The reviewer suggests that we test the Pan-African graph constructed from 1000 Genomes dataset on an independent set of African samples that are not captured by 1000 Genomes dataset. In order to test the generalizability and the effectiveness of the Pan-African graph reference constructed in this study, we measured its performance on a set of independent African samples from the Human Genome Diversity Project (HGDP). This dataset contains samples from 7 African population groups (Mandenka, Yoruba, Biaka, Mbuti, San, Bantu/Kenya and Bantu/South Africa), only one of which is captured in the 1000 Genomes dataset. The

results below (c.f. Figure 3&4 in the main text) show that the Pan-African graph constitutes a more accurate representation of the African populations in the HGDP dataset compared to the Pan-Genome graph or the linear reference GRCh38, and therefore leads to more accurate alignment and more sensitive variant calling. These results support the hypothesis that, even though 6 out of 7 African populations were not explicitly used to construct the Pan-African graph, we were able to capture the shared genetic background through other samples of African ancestry and public databases. We have expanded the manuscript to discuss this important topic raised by the reviewer and provided detailed results in the supplementary material.

Comment #5

- Closely, related to the above issue of sample selection, the paper here relies on comparing people with known ancestry. In the real world, only a fraction of people can claim such homogenous ancestry. Further, ethnic and cultural identity and genetics are known to have a complex and hard to predict relationship. The authors should discuss this issue. Further, how would the authors propose to deal with recently admixed individuals? It feels like the solution proposed directly here would only work for a fraction of people, and this limitation should be discussed.

The solution proposed in the paper works for any group of people as long as the graph reference is constructed with the goal of capturing the genetic background of that group. The individuals in the group do not necessarily have to belong to well-defined ancestry. We still obtain better graph representativeness on the Admixed-American ancestry compared to the Pan-genome graph for example (as mentioned in our response to comment #4). Indeed, the new results we obtained with the gnomad_pangenome graph (shared in response to comment #3 above) proves this point: The gnomad_pangenome graph is an African+European graph, which works better than a pan-genome graph because it still captures a good portion of the African populations. In other words, the African+European graph appeals to both African and European populations and is a better choice compared to the pan-genome graph. However, it is not as good as having a more tailored graph such as Pan-African graphs. The main purpose of constructing a graph reference is to increase representativeness compared to the GRCh38 linear reference, which hardly represents any group of people. As a side note, our solution works also for non-ancestry based populations, e.g. a disease cohort sharing the same genetic cause (unpublished data).

Additionally, as discussed in comment #4, an underperformance can easily be detected in a graph-based approach by monitoring the relevant metrics and a decision can be made whether to use the pangenome graph or augment the existing specific graph.

Comment #6

- We think the AF-based metric makes sense for evaluating how well the graph built matches the graph that was intended to be built. But the "AF_GRAF > AF_GATK" statistics, which the authors try to use to show increased sensitivity aren't good metrics, because they're treating every positive as a true positive: the authors can not know what the real AF actually is, so they can't say identifying more copies of the allele means their tool is more sensitive.

In our measurements, sensitivity measures the detection rate of variants within the cohort. We do not claim those variants to be true positives. At large scales and with whole genome samples (e.g. 1000GP, gnomAD, UK Biobank, Genomics England, Million Veteran Program, etc.), nobody knows the true positive rate, or whether any given variant is a true or not. That is why the commonly-used scientific approach is to use statistical metrics such as Ti/Tv and het/hom ratios and observe if there are any deviations from the expected values. Moreover, identifying an allele with a higher frequency in a population decreases the likelihood of being a false positive due to the unlikeliness of systematic and consistent errors at the same exact locus and the same allele on a large set of samples (especially with Illumina sequencing reads with a random error profile). For example, the commonly used best practice tool, GATK joint calling (recommended for large cohorts), is based on this principle.

Furthermore, it is a common practice for bioinformatics tool developers to support the accuracy measurements at the large scale with more extensive benchmarking on a smaller set of samples such as the Genome in a Bottle datasets, which we have done in our previous work.

The performance of our pipeline was recently re-validated in the precisionFDA's Truth Challenge v2, where it was named as one of the top performers^{1,2} (please see the MHC-Illumina results, the pipeline used therein is the Pan-genome graph used in this study).

¹ <https://precision.fda.gov/challenges/10/results> (retrieved on Jan 11, 2022)

² <https://www.biorxiv.org/content/10.1101/2020.11.13.380741v4.abstract>

Minor points:

Comment #7

- Their title brings in a concept of "the Pan-African genome" which is never mentioned again; Do they mean "African Pan-Genome" or "Pan-African graph" maybe?

We think that as long as we use consistent nomenclature within the manuscript, it is perfectly fine to use the Pan-African genome. Pan-African genome is not an uncommon term and has been used in the literature previously^{1,2}

¹ <https://www.nature.com/articles/s41592-019-0317-y>

² <https://www.science.org/doi/10.1126/sciadv.aaz7835>

Comment #8

- We see where the authors are coming from with the AF accuracy to tell how much a subsample represents the population. But it seems pretty obvious, and not worth so much main text emphasis. Likewise for the diversity / divergence discussion. They say they are "fundamentally independent" but high diversity will generally imply high divergence. (Notably, the paper does not mention that the majority contributor to the existing human reference is a mixrace African-American individual)

We disagree that this is obvious to everyone as it is to the reviewer. We see value in putting emphasis around AF, divergence and diversity. High diversity might imply high divergence in some cases but the reverse statement or the negative statements are not true. We see value in emphasizing these points in the paper.

Since GRCh38 is 70% coming from a single individual¹, it does not constitute any meaningful population representativeness. E.g. one study found that GRCh38 is missing 300M base pairs when compared to African populations². Similar studies on other populations also found a significant amount of missing sequences.

¹ <https://www.ncbi.nlm.nih.gov/grc/help/faq/> (retrieved on Jan 11, 2022)

² <https://doi.org/10.1038/s41588-018-0273-y>

Comment #9

- The authors come up with 651 samples, but link to a data set with 893 AFR samples, and 703 AFR samples from populations collected in Africa. Please explain the discrepancy.

The number of samples used in this study is 661, not 651. As explained in our answer to comment #4, we are not using the related samples in our analyses (the trios in the 1kG dataset). Leaving out the related samples leaves 661 individuals of African ancestry and therefore there is no discrepancy. The reviewer can verify this information in the relevant publication¹.

¹ <https://www.biorxiv.org/content/10.1101/2021.02.06.430068v2>

Comment #10

- The title brings in a concept of "the Pan-African genome" which is never mentioned again; Do they mean "African Pan-Genome" or "Pan-African graph" maybe?

This comment is addressed in our answer to comment #7.

Reviewer #2 (Expertise: Genomics, High performance computing, H3Africa):

The paper manuscript details the generation of increasingly refined graph references to demonstrate population-specific graph references capture more variant information than standard methods such as the GATK/BWA 'best practices' approaches. This include a demonstration that graph references show promise in effectively replacing both joint genotyping and VQSR filtering. In particular, the inclusion of defined metrics to evaluate how particular populations can be assessed is a strong asset that could potentially be applied to other similar analyses in the field.

The manuscript is well-written and analyses are very thorough, and I find the work demonstrates significant promise for the field in pointing to a potential path towards more accurate and sensitive variant calling. I do have some points that should be addressed, though I think they are fairly minor hopefully easily addressable; these are largely outlined below. I am attaching also my notes from the PDF version of the main manuscript which has associated comments.

We thank the reviewer for their valuable comments. We addressed all questions and concerns one by one below, in green color. Where applicable, we have noted the changes we made to the manuscript in light of the reviewer's suggestions.

Comment #1

* I may have misunderstood this point so apologies in advance, but I couldn't find the explanation in the text. In the supplemental methods, specific SVs (insertions) were removed due to the length of the insertion or its similarity to other regions of the genome, thus increasing the potential for ambiguously mapped reads. In addition the multimap filter further prunes the graph based on current haplotypes and mapping. However, one could argue by leaving these out this may lead to some ascertainment bias while tailoring the graph to a population: short reads that *should* be detected as multi-mapping to a graph won't since the offending insertions are filtered out, so perhaps including these regions and leaving these as multi mapped reads is for the best so they can be detected and assessed accordingly. Can you address this point?

It is correct that the main purpose of SV and multimap filters is to resolve ambiguity in the graph reference that one may unwillingly introduce during construction. Indeed, a similar approach called "masking" has been a common practice to ensure proper functionality of bioinformatics toolkits with the reference genomes¹ and recommended by the Genome Reference Consortium² (GRC) and the Genome in a Bottle consortium³. The main justification for this choice is that the useful information lost due to these ambiguities (i.e. due to keeping similar/identical sequences in the reference) is much higher than the misinformation or bias introduced by removing these sequences or the insertions in the graph reference³. It is true that, theoretically, there could be merit in keeping all sequences in the reference and thus leaving ambiguous alignments as they are, for the exact reason pointed out by the reviewer. However, in practice, the most detrimental

cause of inaccurate variant calls in similar regions of the genome is the short read length which cannot resolve these regions and consequently leads to a high number of false negatives (missing variants). So, the current trade-off favors being able to analyze at least one copy of a sequence rather than none, maximizing information with a cost of introducing bias. For any downstream application relying on variant calls, the genomics scientists are highly advised to take caution in interpreting variants in these regions. An important example is the segmental duplications between X and Y chromosomes, which often require not just caution but also some post-processing before being subjected to further analysis and interpretation.

1

<https://gatk.broadinstitute.org/hc/en-us/articles/360041155232-Reference-Genome-Components> (retrieved on Jan 5, 2022)

² <https://www.ncbi.nlm.nih.gov/grc/help/faq/#format-reference-data-for-read-alignment> (retrieved on Jan 5, 2022)

³ <https://doi.org/10.1101/2021.06.07.444885>

Comment #2

* The alignment rates noted and the methods within the main text don't mention the methods used for alignment to the graph in the main text, but it appears this was attained using the GRAF workflow (per the supplemental). It's worth noting this at least once in the main text.

We draw the reviewers attention to the first paragraph of the Results section, which details the alignment and variant calling method used, i.e. Seven Bridges GRAF, with a citation to our original publication around the toolkit. We further mention the best practice pipeline we use as the benchmark, namely BWA+GATK, in the last paragraph of the Results section.

Comment #3

* It should be clarified early on in the main text (intro) that reference genome graphs are generated from already established variant information (e.g. not from de novo variant calls). This *is* mentioned in the methods and in some of the figures, but I think this would best be made explicit early on in the intro to make this clear up-front

Graphs are actually constructed from both available variant calls and do novo variant calls from the cohort under study. At the beginning of the Results section immediately after Introduction (the first 3 paragraphs in the Results section), we mention the following:

1. Population-specific graphs are initially generated from public databases (graph type 2).
2. They are then augmented with cohort specific information. These variants are indeed de novo variant calls produced by the GRAF workflow (graph type 3).
3. We also list the exact public databases used in the construction of the graph references we use with associated citations (e.g. gnomAD, 1000 Genomes and Human Genome Structural Variation Consortium).

Since we kept the Introduction section brief and swiftly moved on to the Results section, we believe that the current information is sufficient to convey the idea that graph references usually (although not mandatory) start with relying on established variant information and then are augmented with cohort-specific de novo calls. However, item 1 above (the point that population-specific graphs we construct start with known variants) also applies to Pan-Genome graphs, which was not clear in the original manuscript. We added a short descriptor to make this clearer.

Comment #4

* Were any general comparisons to other graph-based variant analysis approaches performed to determine whether the same trends exist, such as comparisons to vg? If not, could you clarify why?

In this work, we aimed to compare the utility of various graph reference types and linear references, and also provide a systematic approach to graph reference construction. Therefore, comparing different graph alignment and variant calling methods was not the focus of our study (we made such a comparison in our previous work¹). Nevertheless, as another popular graph-based toolkit, we did conduct a few experiments to measure the performance of VG, which we are happy to share here with the reviewer. Hopefully, the reviewer will agree that even though the graph construction approach we propose in this study can, in principle, be applied to any graph-based bioinformatics toolkit, it is not yet practical to do so in the case of VG due to the current developmental stage of the toolkit.

To measure the accuracy (precision and recall) and computational efficiency, we conducted tests using all five Genome in a Bottle benchmarking samples (HG001 to HG005). In terms of computational efficiency and scalability, VG is unfortunately not yet on par with the state-of-the-art (e.g. BWA+GATK or Seven Bridges GRAF). On a large scale, the computational costs become prohibitively high and graph references containing a high number of variants, i.e. a high number of possible haplotype paths, often leads to execution failure without a significant amount of pruning (a recent example can be seen here²). Computation times separately for graph construction, alignment and variant calling can be seen in the figure below (costs are based on the instances used on AWS). VG has two different aligners: MAP and GIRAFFE. VG GIRAFFE, which is a very recent release, runs much faster than the more accurate algorithm used in VG MAP.

Averaged duration and cost per genome

The runtime for VG GIRAFFE + VG CALL does not appear to be prohibitively high; however, we have concerns about the reliability of the variant calls. To measure the precision and recall, we conducted tests using all five Genome in a Bottle benchmarking samples (HG001 to HG005). As seen in the figure below, both the precision and the recall are so low that it is clear the variant calls produced by VG pipelines are not usable.

Small variant benchmark with HG001-5 (averaged)

Similarly, when we look at the structural variant genotyping performance (below), VG pipelines have a significantly low recall, implying that many of the true positive SVs are missed.

Structural variation benchmark with HG002

Finally, we also look at the Mendelian discordance rate in the GIAB trio HG002-3-4 (please see the figure below), as an orthogonal measure of accuracy (the lower, the better). Again, we observe unsatisfactory performance for VG with discordance rates higher than what the state-of-the-art can accomplish.

Mendelian discordance with Ashkenazim trio

It appears that the graph-aware variant caller VG CALL is under active development and needs further refinements before its performance becomes comparable. It is possible to use one of VG's graph-aware aligners with another variant caller such as GATK HaplotypeCaller or DeepVariant. However, in such cases, the variant calling algorithms cannot utilize the graph reference or the information in graph alignments, and therefore it becomes difficult to call these

graph-based solutions. For example, a common limitation of such pipelines is that they are unable to provide INDEL and SV calling abilities that one expects from a graph-based pipeline.

¹ <https://doi.org/10.1038/s41588-018-0316-4>

² <https://doi.org/10.1186/s13059-021-02474-0>

Comment #5

* The methods suggest that SVs from 1000g PacBio data were added *after* the construction of PanAfrican genome 0. It is noted that the jump from PAG 0 to PAG1 is likely due to this, but is there any reasoning these SVs weren't added in the initial graph construction step (PAG 0)?

In this study, we aim to compare three types of graph references that represent distinct conceptualizations and also have relevance to practical scenarios. To achieve this, we split the human pan-genome into three groups and constructed graphs for each, at each step making the graph more tailored to the cohort under study:

1. Pan-genome: All human populations
2. Pan-African: All African populations
3. Cohort: A graph tailored to the specific cohort

Please see the visualization below.

For Pan-African 0 graph, we are completely relying on public databases without intentionally adding any de novo calls from our test cohort. We think this has practical relevance (1) for developing a general Pan-African graph that can act as a starting point for African population studies (2) for cases where cohort specific variants are not yet available and one must start with the best available graph. All of the graphs in the figure above contain some SVs, whose effect on the results can be seen as an increase in the number of SVs called, from bwa+gatk to Pan-Genome and from Pan-Genome to Pan-African 0 (Figure 4 panel E in the manuscript). Starting with Pan-African 1 graph, we begin adding de novo variants directly from the cohort towards which we are tailoring our graph. The SVs (from pacbio data) that are explicitly added to the Pan-African 1 graph directly come from a subset of the cohort samples; therefore, it

makes sense to add them only to the cohort specific graph. These pacbio SVs do not constitute a reliable source for the African population in general. For instance, Pan-African 0 contains SVs coming from the gnomAD database which has a much larger coverage of African populations.

Comment #6

* The breakdown of the numbers here is a little confusing; can this be clarified? For example, what about the 22% of variants not recovered compared to joint calling?

> (pg 5) The Pan-African 5 graph is able to genotype almost 78% of the variants recovered by traditional joint calling (11,763,827 out of 15,088,205 genotypes across all benchmarking samples), without the need of a post-processing step, while calling less than 18% of the variants filtered out by VQSR (1,625,108 out of 9,049,451 genotypes), providing both sensitivity and specificity.

We agree with the reviewer that a clarification is needed to better explain the differences between the variants called and variants not recovered. To this end, we decided to calculate the quality distributions of these variants. Supplementary Section S4.2 has now been expanded to include these results and a brief discussion. We also refer to these results in the main text where relevant.

To summarize the results, we find that the variants not recovered by the GRAF workflow (the 22%) are those with lower quality and more likely to be false positives. Interestingly, VQSR does not filter out many of these low quality variants. Similarly, the variants that are called by the GRAF workflow but filtered out by VQSR in the GATK workflow contain high quality variants. These observations are more apparent for INDELS, which are more difficult to call compared to SNPs. It is also reaffirming that the portion of variants that, on average, have the highest qualities are called by both GATK joint calling and GRAF pipelines and pass the VQSR filtration. In the calculation of these distributions, we used the genotype quality scores (GQ) and likelihoods (PL) assigned by GATK HaplotypeCaller (and not GRAF quality scores) to avoid circular logic and confirmation bias. Please notice that PL is a better metric to make observations since GQ annotation has an upper cutoff at 99.

Comment #7

* Do you recommend using the approach for graph generation in the manuscript, or is this meant to be more of a proof-of-concept demonstrating that pop-specific graphs are much better at capturing variation? For example, IMO the graph construction methods seem very elaborate, suggesting that regeneration of a graph based on varying the AF cutoff, length and remapping filtering of SVs, etc could be limiting factors in using this approach; re-generating graphs could be computationally challenging and expensive if filtering steps or criteria for inclusion of samples are adjusted, whereas a more comprehensive/inclusive approach that can be post-filtered (or masked) ad hoc might be better.

We hope to achieve both with our manuscript; recommend the proposed graph construction approach as an effective means that can be used for graph-based bioinformatics, and also show that more tailored references, e.g. population-specific, are much better than generic ones. The experiments on the African populations are meant to demonstrate the utility of this approach in the population-specific context. However, we believe that this concept can be extended to other cohort definitions such a disease-specific cohort, or a pedigree graph reference, both of which we are actively working on.

We agree with the reviewer that the manuscript lacks a section transparently discussing the computational needs for the proposed graph construction approach. We already had a supplementary section discussing the computational efficiency measurements (Supplementary Section S5). We now expanded this section to explicitly include graph construction times.

Contrary to how it may appear at first sight, the construction procedure is not intensive and can be easily repeated with each change in the graph reference (or the sample composition). For all graphs constructed in this study, our algorithms take between 2-3 hours on 72 CPUs and use around 16-17GB of memory. Some of the steps in our algorithms are currently single-threaded and therefore we expect the construction time to be much shorter than the current value in the near future. It is true that the construction pipeline contains parameters that may need to be tuned depending on, for example, the choice of AF cutoff, the sequencing read length or the diversity of the population. We aimed to design a construction approach that is lightweight enough to allow experimentation and tuning during construction. Please also find the computational metrics below.

Graph name	Graph construction	Graph and index loading time	Graph and index memory usage	Alignment time	Variant call time
PanGenome	1h 52m	164s	16.70 GB	2h 42m	41m
Pan-African 0	1h 34m	154s	16.01 GB	2h 54m	41m
Pan-African 1	2h 23m	164s	16.43 GB	4h 21m	46m
Pan-African 2	2h 41m	164s	16.71 GB	4h 31m	47m
Pan-African 3	2h 47m	173s	16.69 GB	4h 38m	47m
Pan-African 4	2h 54m	173s	16.73 GB	4h 41m	47m
Pan-African 5	3h 01m	174s	16.85 GB	4h 43m	47m

Table 1. Execution details for graph, alignment and variant calling

An all-inclusive approach that is subsequently filtered down depending on the population is an interesting idea; however, currently we can not think of a way to achieve this easily since addition/subtraction of even a single edge/variant might change the possible haplotypes paths significantly and all filtering steps (SV, multimap, etc) needs to be applied from scratch, voiding computational benefit of this approach. Nevertheless, this may be a future improvement of our methodology.

Reviewer #3 (Expertise: Genomics, H3Africa):

Population-specific genome graphs improve high-throughput sequencing data analysis: A case study on the Pan-African genome

General comments

Authors have addressed an important question: The bioinformatic methods i.e. Genome Graphs, to improved genome analysis in order to alleviate the disparity in populations variants representativity in the current reference genomes, with a specific focus on African populations.

The study is well designed, executed, and written to accommodate a wide range of specialist and non-specialist readers. The article provides some evidence confirming or supporting that :
1- At the genomic level, African populations is both more diverse and divergent from the current reference genome; 2- More diverse the population is, higher is the number of samples needed to read an appropriate level of representativity; 3- Population specific progressive graph mapping performed much better, compare to pan genome approach; 4- tailoring genome graph to specific cohorts improved graph construction, aliment and variant calling outcomes; 5- the higher the number of informative read, and the lower the number of alignment error rate....
The article provide data suggesting the Genome graphs approach will add, and likely improve the currently available bioinformatics tools kit, while improving analytics for all populations, and particularly the underrepresented and more diverse African populations.

While the article will add to the global literature, it could be improved in a few areas.

We thank the reviewer for their valuable comments. We have addressed all questions and concerns one by one below, in green color. Where applicable, we have noted the changes we made to the manuscript in light of the reviewer's suggestions.

Comment #1

Specific comments

1- Regarding Accuracy, Efficiency, and Applicability: Can the author uses their approach to compare/ replicate with in more diverse set of African genome data, such the one recently reported by the H3Africa Consortium?

Nature, 2020 Oct;586(7831):741-748; doi: 10.1038/s41586-020-2859-7. PMID: PMC7759466

Specifically, using the H3Africa genome data, to explore comparatively, using Genome Graphs:

- capturing the number of novel SNV...
- investigating the proportions o of actionable genes variants
- numbering of novel SNVs?
- Etc.....

We thank the reviewer for pointing out a seminal paper on the African genome. We are indeed aware of this valuable study and the dataset comprising WGS data from many African

populations. The goal of our work has been to establish a systematic approach to graph reference construction and show the utility of using population-specific genome graphs. We think that this is best achieved through the validation of the methodology on well-established, extensively studied and widely accessible datasets such as 1000 Genomes project and the gnomAD database, both of which are publicly available without any restrictions and therefore enable comparison and verification of our results. We hope that our study will pave the way for novel discoveries on both old (e.g. 1000 genomes) and new (e.g. H3Africa) datasets by extracting more insights from the raw sequencing data.

H3Africa constitutes an invaluable resource that leads (and will continue to lead) to many important discoveries in the field of genomics. We agree that this dataset would be an excellent opportunity to extend our work and measure, in more detail, the utility of GRAF in tertiary analyses. However, given that H3Africa requires an access request to be reviewed by the DBAC committee and that its priority is to focus on health (e.g. disease studies), we believe that such a study is best conducted with an active collaboration from all relevant participants. We are hoping that in the future as a follow-up study we will be able to apply the proposed method in our manuscript to H3Africa dataset.

Comment #2

2- How good can the Genomes Graphs method better characterizes the so-call “Ghost DNA” in African genomes? Can the author provide some evidence?

We believe that this important question is unfortunately out of scope for this study. The graph-based approaches should do better in capturing any variation that may be shared among the samples of a given population; however we do not expect to see any tendency towards higher sensitivity for ghost DNA sequences compared to other non-ghost DNA sequences that are still under-represented in the current human genome reference.

Comment #3

3- Can the approach be used to identified marker of natural selection, known loci such as Sickle cell disease mutation, alpha-thalassemia, G6PD, APOL 1 variant, or other novel loci under natural selection, that expected to be more common in African genomes?

Our graph approach would be more capable of genotyping those variants at the population level compared to the approaches relying on linear genome references. Moreover, we expect, through sensitive variant calling enabled the graph approach, there is a better chance for identifying/discovering genetic mechanisms for not only single gene/locus diseases such as the ones the reviewer mentioned but also diseases with complex etiologies. However, similar to the reviewer’s previous comment (#2), we believe this falls outside the scope of our work and would require more targeted cohorts.

Comment #4

4- Can the authors comment on the Number of African genomes needed to accurately capture the variations African populations?

We comment on this topic in Figure 2 panel B in the main text, Figure S7 in the supplementary material (please also see Supplementary Section S2), and Supplementary Table S1 (in the excel document). We also briefly discuss this topic in the first paragraph of the section titled “Population-Specific Graph Construction” and also in the second paragraph of the Discussion section. To summarize, as one would probably expect, there is no single answer to the reviewer’s question. However, we can estimate the true positive and false positive rate in a population-specific graph reference; these rates indicate how well a population is captured (Figure 2 and S7). Based on our measurements on the 1000 Genomes dataset (Table S1), the number of samples can range from hundreds to thousands depending on the desired level of representativeness. Nevertheless, even a few tens of samples can go a long way in improving variant detection compared to the linear reference approach. In Table S1, we also provide numbers of samples required for European, East Asian, South Asian and Admixed-American populations. As expected, the African population requires the highest number of samples to attain the same level of representativeness compared to other populations because it is the most diverse population.

Inevitably, as new and more diverse datasets become available, these estimations are bound to change due to novel variant discoveries. For this exact reason, our graph construction approach allows for augmentation with backwards and forwards compatibility.

Reviewers' Comments:

Reviewer #1:

Remarks to the Author:

Overall we are pleased with the prior response to reviews. However, we'd like to make the following points:

- We are pleased that you now compare a population-specific graph (Pan-African 0) with a broadly equivalent non-specific graph (Pan-Genome-gnomAD, in figures S8 and S9), but to us the results look almost identical between the two, on most of the metrics, and you don't really do any statistical tests to tease them apart. You do say that gnomAD has more African-ancestry samples in it than it ought to to really make a non-population-biased control. But then if we accept that as an explanation for why it looks so much like the matched experimental condition, you're back to not having a negative control.
- There are still have no negative controls for the HGSVC and cohort-augmented-graphs (Pan-African 1 and Pan-African 2-5). It is okay dismiss our suggestions for how a negative control might be made, but then you really need to come up with something to replace them!
- We appreciate that you ran on some additional African populations to prove that you are getting a graph with good coverage for what African-ancestry people are likely to have in their genomes. However, you don't do the experiments needed to relate that causally to the matching of the graph construction and test set super-populations. For all we know, the Pan-African 5 graph you end up with might work equally great across all populations. In fact, because most common variation has its ancestral origin in Africa we might plausibly expect the graph to work for most human populations. This would be less of a problem if you only pitched your method as a great way to augment your graph on-line with samples from your cohort. If you shuffle your cohort, it sort of definitionally has allele frequencies, and you can use them as priors. You prove that if you augment with your cohort it beats not doing that. But you're pitching this as "population-specific" and not "cohort-specific". As evidenced by your additional African test populations, you're asserting you have a graph that captures something about the AFR super-pop specifically. That might be true! Super-pops also have allele frequencies! But essentially you don't really prove you're capturing anything about the super-pop.
- You also still aren't using the right words for populations and would urge you check carefully language describing populations to avoid inadvertent generalization or misspecification; we still have "African super-population", without "ancestry", and in figure S10 you've ignored all the advice for the different sub-populations. For example, it is important to refer to that community as "Gujarati Indians in Houston, Texas, USA" when describing these samples in articles or presentations. Including the full name reinforces the point that the sample set does not represent all Gujarati people, whose population history is complex.

Reviewer #2:

Remarks to the Author:

I found questions raised by other reviewers to be very pertinent though I leave it to their discretion as to whether the authors addressed them accordingly. As for myself, I found all questions I had to be adequately answered; in addition I thank them for pointing out one particular point I missed leading to one question. Best of luck!

Reviewer #3:

Remarks to the Author:

The authors have adequately addressed our comments.

This article have properly investigated, and provides an innovative approach for Genomes analysis, an important contribution to the global literature.

Reviewer #1 (Remarks to the Author):

Overall we are pleased with the prior response to reviews. However, we'd like to make the following points:

Comment #1

- We are pleased that you now compare a population-specific graph (Pan-African 0) with a broadly equivalent non-specific graph (Pan-Genome-gnomAD, in figures S8 and S9), but to us the results look almost identical between the two, on most of the metrics, and you don't really do any statistical tests to tease them apart. You do say that gnomAD has more African-ancestry samples in it than it ought to to really make a non-population-biased control. But then if we accept that as an explanation for why it looks so much like the matched experimental condition, you're back to not having a negative control.

We compared the Pan-Genome-gnomAD results to other graphs using the Wilcoxon test and the difference is significant (except for SV counts to Pan-African-0). We realized that these significance tests were missing in the text; therefore, we now added them to the supplementary document under relevant figures. Although distributions are statistically different enough, the nature of gnomAD data provides a certain bias towards African and European samples (please also see our response to comment #3 below). Therefore, we do not claim that it is a proper negative control. In order to eliminate any bias towards a population, one would need to start with datasets that have uniform sample distributions across populations. Our Pan-Genome graph was constructed with these constraints, therefore we respectfully disagree with the reviewer and we think it is the most appropriate negative control. Additional clarifications are given in our response to Comment #3.

Comment #2

- There are still have no negative controls for the HGSVC and cohort-augmented-graphs (Pan-African 1 and Pan-African 2-5). It is okay dismiss our suggestions for how a negative control might be made, but then you really need to come up with something to replace them!

We thank the reviewer for their comment but we respectfully disagree with the reviewer's framing of negative control in our experiments. We arranged the construction and benchmark groups to specifically address the negative control that is needed at each step. HGSVC data is collected from 1000 Genome samples therefore it is part of the cohort used in our study and does not pertain to the general African super-population. Therefore, it was added at the first step of cohort-specific graph construction (Pan-African-1) along with other cohort information. The initial Pan-African-0 graph that was constructed from publicly available data is the control for that step. Similarly each iteration in the Pan-African 0-5 series acts as a control for the subsequent iteration. For any cohort-specific graph (Pan-African 1-5), a pan-genome counterpart/control is irrelevant.

Comment #3

- We appreciate that you ran on some additional African populations to prove that you are getting a graph with good coverage for what African-ancestry people are likely to have in their genomes. However, you don't do the experiments needed to relate that causally to the matching of the graph construction and test set super-populations. For all we know, the Pan-African 5 graph you end up with might work equally great across all populations. In fact, because most common variation has its ancestral origin in Africa we might plausibly expect the graph to work for most human populations. This would be less of a problem if you only pitched your method as a great way to augment your graph on-line with samples from your cohort. If you shuffle your cohort, it sort of definitionally has allele frequencies, and you can use them as priors. You prove that if you augment with your cohort it beats not doing that. But you're pitching this as "population-specific" and not "cohort-specific". As evidenced by your additional African test populations, you're asserting you have a graph that captures something about the AFR super-pop specifically. That might be true! Super-pops also have allele frequencies! But essentially you don't really prove you're capturing anything about the super-pop.

We agree that the AFR graph would contain considerable common variation with the more general Pan-Genome graph (see comparison below).

However, it also includes many AFR-specific common variants that are absent (or observed at low frequencies) in other populations, and excludes common variations from other populations that are not relevant to AFR. The influence of this difference is apparent in the truth set based benchmarking (see below) of GIAB (Genome-in-a-Bottle) samples HG001, HG002 (European ancestry) and HG005 (Chinese ancestry), where we measure the F1-score for variant calling using the Pan-Genome, gnomAD Pan-Genome and Pan-African-5 graphs. Pan-African-5 graph, being more tailored towards African ancestry samples, suffers considerably in accuracy for non-African samples. It also important to observe that gnomAD Pan-Genome performs slightly better than Pan-Genome in HG001 and HG002 because of the bias towards European ancestry in its sample distribution, and performance degrades for HG005 due to the underrepresentation of East Asian ancestry in gnomAD database. And this is precisely the reason why we chose not to use gnomAD Pan-Genome as our control graph.

We also would like to reiterate that we use a more relaxed definition for the term “population”: A collection of individuals with some putatively shared genomic background. The level of shared background would depend on the study and we argue that it is beneficial to tailor the reference graph used for analysis. Since our cohort in this study represents a broad collection of samples from 1000 Genomes African ancestry super-population, it would be more representative of another African ancestry sample, as in the case of HGDP AFR benchmarks. Even though our construction cohort does not completely overlap with the HGDP samples, Pan-African graph provides a better base than Pan-Genome graph.

Comment #4

- You also still aren't using the right words for populations and would urge you check carefully language describing populations to avoid inadvertent generalization or misspecification; we still have "African super-population", without "ancestry", and in figure S10 you've ignored all the advice for the different sub-populations. For example, it is important to refer to that community as "Gujarati Indians in Houston, Texas, USA" when describing these samples in articles or presentations. Including the full name reinforces the point that the sample set does not represent all Gujarati people, whose population history is complex.

We explicitly define what is meant when we use African super-population, populations or samples in the paper: *“the 7 populations of African ancestry (ACB, ASW, ESN, GWD, LWK, MSL, YRI) in the 1000 Genomes dataset”*. Our conclusions follow from this sample set, so it is clear what we mean when we refer to “African super-population” later in the text. For S10, we opted to use the shorter population names provided by 1000 Genomes metadata for brevity and readability. We thankfully acknowledge the point made by the reviewer to better describe the populations, therefore we replaced the figure with full descriptions instead.

Reviewer #2 (Remarks to the Author):

I found questions raised by other reviewers to be very pertinent though I leave it to their discretion as to whether the authors addressed them accordingly. As for myself, I found all questions I had to be adequately answered; in addition I thank them for pointing out one particular point I missed leading to one question. Best of luck!

We thank the reviewer for their valuable comments.

Reviewer #3 (Remarks to the Author):

The authors have adequately addressed our comments. This article have properly investigated, and provides an innovative approach for Genomes analysis, an important contribution to the global literature.

We thank the reviewer for their valuable comments.

Reviewers' Comments:

Reviewer #1:

Remarks to the Author:

Thanks, the authors have adequately addressed our concerns.